# FOCUS & RePAIR: Mitigating Text Degeneration via Token-Level Guidance for Pruned Large Language Models

**Junyoung Lee** [1]  **Sehyeon Park** [2]  **Shinhyoung Jang** [2]  **Seonha Ryu** [2]  **Hojeong Kim** [2]  **Hyunsei Lee** [2]  **Il hong Suh** [3]  **Yeseong Kim** [1]

## Abstract

Pruning is a practical approach to compress large language models (LLMs), but it can amplify text degeneration, especially repetition loops, even when perplexity and task accuracy remain largely unchanged. In this work, we present a token-level analysis of this failure mode by viewing decoding as a dynamical process that enters and persists in a small set of recurrent contexts. Our analysis decomposes degeneration into *loop entry risk* and *loop persistence*, and shows that persistence is controlled by the *escape mass* assigned to plausible alternatives within the token sampling set. Motivated by these findings, we propose two token-level guidance objectives for post-pruning fine-tuning. FOCUS reweights distillation toward high-confidence teacher regions to suppress leakage, while RePAIR uses onset-centered positive/negative continuation pairs with a margin loss to promote plausible alternatives and prevent early commitment to repetition loops. Experiments on open-ended continuation and instruction-based generation show that both methods consistently reduce repetition and improve generation quality.

## 1. Introduction

Large language models (LLMs) have become foundational in a broad spectrum of applications, ranging from dialogue systems and code assistants to knowledge-intensive question-answering and content generation (Minaee et al., 2025; Jiang et al., 2024; Yue, 2025). In practical deploy-

*Table 1.* Comparison of repetition rates between unpruned and pruned models. The pruned model is finetuned on the Alpaca dataset. For decoding, we employ top-$p$ sampling with $p = 0.9$.

| Condition | Sampling | Greedy |
|---|---|---|
| Unpruned | 5.9% | 26.6% |
| Width pruned | 12.4% | 63.1% |
| Depth pruned | 15.4% | 63.7% |

ments, the execution of large models is far from straightforward: Inference latency escalates dramatically under constrained serving budgets, and memory requirements often exceed the limits of commodity hardware (Chitty-Venkata et al., 2024; Pope et al., 2022). To manage these constraints, prior research has developed pruning strategies, ranging from depth pruning, which eliminates entire transformer blocks, to width pruning, which removes internal submodules such as attention heads or MLP channels (Ma et al., 2023; Kim et al., 2024; Frantar & Alistarh, 2023; Lee et al., 2025). Because pruning inevitably discards parameters that encode useful behaviors, practitioners typically perform a post-pruning finetuning stage to restore degraded capabilities. For example, these techniques recover knowledge in pruning settings with a relatively small dataset, e.g., the Alpaca dataset with its instruction–response pairs (Taori et al., 2023).

Most prior work on pruning has focused on knowledge preservation (Li et al., 2024; Park et al., 2024). Standard evaluations assess whether a pruned model preserves perplexity, zero-shot accuracy, and downstream task performance relative to its unpruned counterpart. However, there is growing empirical evidence that pruning can also introduce undesirable side effects, *even when* principal metrics such as perplexity or accuracy appear to be intact (Liebenwein et al., 2021; Jordao & Pedrini, 2021; Jaiswal et al., 2024). A primary issue is text degeneration, where the model **repeatedly generates** the same words or phrases, for example: "<prefix> ... for the deceased. The cemetery is designed to be a peaceful place. The cemetery is designed to be a peaceful place. The cemetery is designed to be a peaceful place. ...". To demonstrate this, we generate 200 tokens from the WikiText-103 dataset using a 50-token prefix

[1]Department of Electrical Engineering and Computer Science, POSTECH, Pohang, Republic of Korea [2]Department of Electrical Engineering and Computer Science, Daegu Gyeongbuk Institute of Science and Technology (DGIST), Daegu, Republic of Korea [3]Hanyang University, Seoul, Republic of Korea. Correspondence to: Junyoung Lee <lolcy3205@postech.ac.kr>, Yeseong Kim <yeseongkim@postech.ac.kr>.

*Proceedings of the 43$^{rd}$ International Conference on Machine Learning*, Seoul, South Korea. PMLR 306, 2026. Copyright 2026 by the author(s).

with both the unpruned Llama model and the pruned model after fine-tuning. A sentence is classified as repetitive if repeated segments account for the majority of the generated text, as further discussed in Section 3. As shown in Table 1, the degeneration phenomenon becomes more severe after pruning in both cases. This observation indicates that, although simple fine-tuning can recover knowledge to some extent, it remains essential to mitigate the side effects that arise during text generation.

Previous studies have noted that text degeneration occurs when previously generated tokens increase the likelihood of the model producing the same tokens again (Holtzman et al., 2020; Welleck et al., 2019; Xu et al., 2022). To address this, these approaches lower the probabilities of previously generated tokens while increasing those of tokens that have not appeared. Although this strategy is effective in reducing repetition, it does not provide guidance on which tokens should be generated to produce a coherent continuation, resulting in degraded perplexity.

In this work, we identify repetition-loop degeneration as a *token-level dynamical phenomenon* induced by the interaction between the next-token distribution and the decoding rule. Using a coverage-based view of repetition, we find that degeneration typically behaves as a sharp entry event: *replacing only a small number of high-probability tokens at the loop onset is often sufficient to steer generation away from a repetitive trajectory*. We formalize this behavior by decomposing repetition into (i) *loop entry risk* and (ii) *loop persistence*, where persistence dominates long-horizon repetition because it compounds over consecutive in-loop transitions. This analysis also explains why pruning and naive distillation can exacerbate degeneration by simultaneously increasing leakage into teacher-suppressed tokens (raising entry risk) and collapsing near-tie alternatives at loop-sensitive contexts (raising persistence).

These findings suggest that mitigating degeneration after pruning requires *training-time* control over token probabilities at loop-sensitive contexts: suppressing leakage toward teacher-suppressed tokens to reduce loop entry, while preserving plausible alternatives to prevent early commitment and reduce loop persistence. We therefore propose two complementary token-level guidance methods, **FOCUS** and **RePAIR**. FOCUS modifies standard distillation by reweighting tokens to emphasize high-confidence teacher regions, discouraging probability mass from drifting into low-support (leakage) areas under capacity constraints. In contrast, RePAIR constructs paired continuations around degeneration onsets, i.e., a repetitive negative sample and a non-degenerate regeneration, and applies a margin-based objective that explicitly promotes plausible alternatives at the onset. We evaluate our proposed method in the post-pruning pipeline by pruning Llama-family models and fine-tuning

with LoRA. Across open-ended continuation (WikiText-103) and instruction-based generation (Self-Instruct), FOCUS and RePAIR consistently reduce repetition and improve distributional and semantic quality (e.g., MAUVE, CREP, $EAD_1$, and BERTScore), while incurring only a small perplexity increase. Moreover, FOCUS is compatible with existing training-based mitigation objectives and improves their generation quality when combined.

We summarize our contributions as follows:

- We introduce a token-level framework for repetition-loop degeneration that separates *loop entry* from *loop persistence*, and derive nucleus-decoding diagnostics that expose how probability allocation among *nucleus-contained* alternatives governs repetition.

- We propose two complementary token-level guidance objectives, FOCUS and RePAIR: FOCUS suppresses distillation-induced leakage by emphasizing high-confidence teacher regions, while RePAIR uses onset-centered positive/negative continuation pairs to promote plausible alternatives at loop-sensitive contexts.

- Across open-ended continuation and instruction-based generation, our methods consistently reduce degeneration and improve generation quality (e.g., MAUVE and $EAD_1$) with only a small perplexity increase.

## 2. Related Work

### 2.1. LLM Pruning & Distillation

Model pruning has been extensively investigated as an effective strategy to reduce model size and computational overhead. However, pruning inevitably introduces knowledge loss and results in performance degradation (Kim et al., 2024; Ma et al., 2023). To address this, knowledge distillation (KD) is commonly employed in the post-pruning fine-tuning stage to transfer knowledge from the teacher to the pruned student model (Xu et al., 2024; Gu et al., 2024), offering the advantage of leveraging soft probabilities to provide richer supervision than one-hot labels. Nevertheless, KD does not always yield consistent benefits (Ma et al., 2021; Zhang et al., 2025). For example, naive KD may transfer incorrect answers from the teacher and they demonstrate that student models trained with appropriate corrections of teacher logits can even outperform the teacher itself in classification tasks (Zhang et al., 2024). Several studies have also noted that naive knowledge distillation may fail to effectively train the student model, either due to the capacity gap between teacher and student or biases inherent in the teacher model (Zhong et al., 2024; Shum et al., 2024). These observations highlight the need for strategies that selectively extract and transfer only useful information from the teacher to improve training effectiveness.

## 2.2. Mitigation of Text Degeneration

Text degeneration has been addressed through both decoding-time and training-time strategies. On the decoding side, deterministic methods such as greedy and beam search often suffer from limited diversity and degeneration. Stochastic alternatives, including top-$k$ (Fan et al., 2018) and top-$p$ (nucleus) sampling (Holtzman et al., 2020), improve diversity by restricting generation to high-probability tokens, with top-$p$ adapting to the distribution's sharpness to produce more natural outputs.

Complementary to decoding-based methods, training-time approaches modify the learning objective to discourage repetition. Unlikelihood training penalizes previously generated tokens by reducing their probabilities (Welleck et al., 2019), while ScaleGrad adjusts token-level gradients to promote novel tokens (Lin et al., 2021). However, unlikelihood-based methods often degrade perplexity. More recently, DITTO trains on synthetic sentence-level repetition data to suppress repeated tokens (Xu et al., 2022).

# 3. Token-level Guidance: Repetition-loop Dynamics and Supervision Signals

This section develops a *token-level account* of repetition-loop degeneration under common decoding rules, e.g., nucleus (top-$p$) decoding and identifies which parts of the next-token distribution must be corrected during training.

## 3.1. Coverage-based Repetition Metrics and Onset Sensitivity

Repetition-loop degeneration typically manifests as a short recurring $N$-gram that explains a large fraction of the generated sequence. To quantify this concentration, let $s_{1:T} = (s_1, \ldots, s_T)$ be a generated token sequence and let $r$ denote an $N$-gram. We define the **Coverage** of $r$ as the fraction of tokens explained by its occurrences:

$$\text{Coverage}(r, s_{1:T}) \triangleq \frac{1}{T} \sum_{j=1}^{T-N+1} \mathbf{1}[s_{j:j+N-1} \approx r] \cdot N, \quad (1)$$

where $\approx$ is an exact match under tokenization (or a task-specific equivalence used in evaluation). Since degeneration concentrates on a small number of patterns, we summarize each sequence by the dominant pattern:

$$\text{Coverage}(s_{1:T}) \triangleq \max_{r \in \mathcal{V}^N} \text{Coverage}(r, s_{1:T}), \quad (2)$$

and report dataset-level degeneration by the **Coverage-based REPetition** rate (CREP),

$$\text{CREP}(D) \triangleq 100 \times \frac{1}{|D|} \sum_{s \in D} \mathbf{1}[\text{Coverage}(s) \geq \tau], \quad (3)$$

*Table 2.* Degeneration rates of the unpruned model before and after correcting two onset tokens.

| Condition | Before | After |
|-----------|--------|-------|
| Sampling  | 5.9%   | 0.7%  |
| Greedy    | 26.6%  | 10.8% |

where $\tau$ is a fixed threshold and $N, \tau$ are held constant unless stated otherwise. Coverage makes repetition measurable as a dominance event: once a short pattern occupies a large portion of the sequence, decoding spends many steps revisiting a small set of local contexts.

We next show that repetition commonly behaves as an *entry event* that can be disrupted by a minimal local intervention. We generate 1,000 sequences, identify degenerated outputs using CREP, and locate the onset of the first repetition loop as the earliest position where the dominant $N$-gram begins to recur and then continues over a contiguous span. At this onset, we modify only the first two tokens by selecting an alternative among the top-2 candidates under the model distribution at the same prefix, then regenerate the continuation and re-evaluate CREP.

Table 2 shows that replacing only two onset tokens reduces CREP from 5.9% to 0.7% under sampling and from 26.6% to 10.8% under greedy decoding. The replacements are restricted to high-probability candidates, so the local continuation remains plausible while the trajectory shifts away from a repetitive basin. This sensitivity indicates that degeneration often depends on a small subset of loop-sensitive contexts near the first onset.

## 3.2. Decoding Dynamics: Entry Risk and Persistence

To connect onset sensitivity to token-level probabilities, we adopt a decoding-dynamics view. For a fixed $N$-gram context representation $c_t \triangleq s_{t-N+1:t-1}$, the model induces a conditional distribution $p_\theta(\cdot \mid c_t)$. Under a fixed decoding rule (for example, top-$p$ sampling), generation induces a stochastic process over contexts because sampling selects a token and the context updates deterministically. Repetition-loop degeneration corresponds to trajectories that enter and then remain within a small recurrent subset of contexts. We denote such a repetition-prone region by $\mathcal{L}$, which can be operationalized as the set of contexts that repeatedly regenerate the dominant $N$-gram identified by Coverage.

This perspective separates degeneration into *loop entry* and *loop persistence*. Let the loop-hitting time be $\tau_{\mathcal{L}} \triangleq \min\{t \geq 1 : c_t \in \mathcal{L}\}$ and define the horizon-$T$ entry risk

$$\mathcal{R}_T \triangleq \mathbb{P}(\tau_{\mathcal{L}} \leq T). \quad (4)$$

Once decoding reaches $\mathcal{L}$, repetition depends on the probability of remaining inside it. For $c \in \mathcal{L}$, define the *one-step*

*persistence probability*

$$\rho(c) \triangleq \mathbb{P}(c_{t+1} \in \mathcal{L} \mid c_t = c), \qquad \bar{\rho} \triangleq \sup_{c \in \mathcal{L}} \rho(c). \quad (5)$$

Coverage-based degeneration requires sustained residence in $\mathcal{L}$. Let $\ell_\tau$ be the minimum number of consecutive in-loop transitions required for $\mathrm{Coverage}(s_{1:T})$ to exceed $\tau$. Then a simple decomposition upper bounds degeneration by

$$\mathbb{P}(\mathrm{Coverage}(s_{1:T}) \geq \tau) \leq \mathcal{R}_T \cdot \bar{\rho}^{\ell_\tau}. \quad (6)$$

Equation (6) yields a structural implication.[1] Loop entry contributes linearly through $\mathcal{R}_T$, whereas persistence is exponentiated by the required run length $\ell_\tau$. As a result, moderate reductions in $\bar{\rho}$ can suppress long repetition runs even when entry events are not fully eliminated. This explains why changing only a few high-probability tokens near onset can produce a large reduction in CREP in Table 2.

### 3.3. Nucleus Alternatives and Escape Mass

The remaining question is which token-level distortions control $\rho(c)$ under nucleus sampling, where only candidates inside the nucleus set are sampleable. Fix a context $c$ and let $S_p(c)$ denote the nucleus set, defined as the smallest set of tokens whose probability mass under $p_\theta(\cdot \mid c)$ is at least $p$. Top-$p$ sampling draws the next token from the renormalized distribution on $S_p(c)$. Therefore, persistence inside a loop region depends on how the *nucleus mass* is divided between loop-continuing tokens and tokens that exit the loop region.

For a loop context $c \in \mathcal{L}$, define the set of loop-continuing nucleus tokens

$$A(c) \triangleq \{v \in S_p(c) : \mathrm{update}(c,v) \in \mathcal{L}\}, \quad (7)$$

where $\mathrm{update}(c,v)$ is the context update after appending token $v$. Let

$$a(c) \triangleq \sum_{v \in A(c)} p_\theta(v \mid c), \qquad e(c) \triangleq \sum_{u \in S_p(c) \setminus A(c)} p_\theta(u \mid c), \quad (8)$$

where $a(c)$ is the unnormalized *loop mass* within the nucleus and $e(c)$ is the *escape mass* within the nucleus. Since nucleus sampling renormalizes over $S_p(c)$, the persistence probability satisfies

$$\rho(c) = \frac{a(c)}{a(c) + e(c)}. \quad (9)$$

This identity makes the control mechanism explicit: *reducing persistence requires increasing escape mass* inside *the nucleus, not merely increasing entropy in the full vocabulary.*

---

[1]Consistent with prior repetition analyses, we observe an increasing and saturating trend in the mean probability of repeated tokens, which suggests a rising tendency toward persistence rather than a direct estimate of $\bar{\rho}$, which is provided in Appendix A.

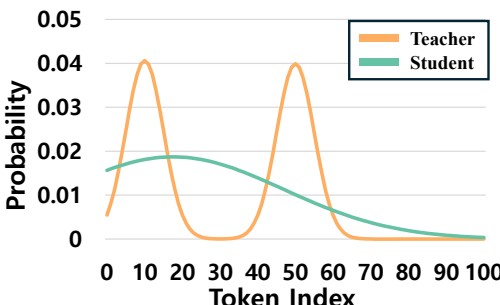

*Figure 1.* Mode averaging under forward KL minimization. The student curve corresponds to the numerically optimized single-mode approximation of a multimodal teacher.

### 3.4. How Pruning and Naive Distillation Reshape Risks

Previous work notes that text degeneration often arises when model uncertainty falls outside a stable entropy range, with both overly confident and overly uncertain distributions (Arora et al., 2023). Building on this observation, we now connect common student distortions after pruning and standard knowledge distillation to the two drivers in (6). Let $q(\cdot \mid c)$ denote the teacher distribution and $p(\cdot \mid c)$ the student distribution. Naive distillation commonly minimizes the forward KL divergence

$$\mathcal{L}_{\mathrm{KL}}(q\|p) = \sum_{v \in \mathcal{V}} q(v \mid c) \log \frac{q(v \mid c)}{p(v \mid c)}. \quad (10)$$

Because each token contributes proportionally to $q(v \mid c)$, tokens with negligible teacher probability receive little direct weight in (10). Under capacity constraints induced by pruning, the student may not represent the teacher's multi-modal structure in a single context. In that case, minimizing (10) can yield a single broadened approximation that compromises across teacher modes, which redistributes probability into intermediate and low-density regions of the teacher distribution. Figure 1 illustrates this mode-averaging behavior in a continuous toy example.

**Tail Leakage Increases Loop Entry Risk.** Define the teacher-suppressed set

$$\mathcal{T}_\epsilon(c) \triangleq \{v \in \mathcal{V} : q(v \mid c) \leq \epsilon\}, \quad (11)$$

and the student's leakage mass $\Delta_\epsilon(c) \triangleq \sum_{v \in \mathcal{T}_\epsilon(c)} p(v \mid c)$. Forward KL provides limited pressure against allocating mass to $\mathcal{T}_\epsilon(c)$, especially when the student must approximate multiple teacher modes with reduced capacity. If leakage mass shifts into tokens that are occasionally admitted into $S_p(c)$, these tokens become sampleable under nucleus decoding and can increase the entry term $\mathcal{R}_T$.

**Alternative Collapse Increases Loop Persistence.** At loop-sensitive contexts near the first onset, teachers often

assign comparable probability to multiple semantically consistent next tokens. Pruning-induced representational loss can distort these near-ties, producing a single dominant continuation and suppressing competing alternatives. Under nucleus sampling, this distortion reduces escape mass $e(c)$ relative to loop mass $a(c)$ in (8), thereby increasing $\rho(c)$ through (9) and inflating $\bar{\rho}$.

Taken together, *pruning and naive distillation can increase both loop entry risk and loop persistence* by (i) allocating non-negligible probability to teacher-suppressed tokens and (ii) collapsing teacher-supported alternatives at loop-sensitive contexts.

These observations translate into two supervision targets that follow directly from the entry-persistence decomposition. First, to reduce *entry risk*, training should suppress leakage toward teacher-suppressed regions so that such tokens do not enter the nucleus set and become sampleable triggers under top-$p$ decoding. Second, to reduce *persistence*, training should preserve and, when necessary, promote *plausible* escape alternatives *within* the nucleus at loop-sensitive contexts, thereby increasing escape mass $e(c)$ and decreasing $\rho(c)$ via (9) rather than injecting diffuse probability into tokens that remain outside $S_p(c)$. Section 4 instantiates these targets with two complementary objectives: one explicitly controls tail leakage relative to the teacher, and the other reallocates probability among onset-adjacent candidates to maintain nucleus-level escape mass and prevent early commitment to repetition loops.

## 4. Method

In this section, we propose the token probability weighted distillation method, which follows the trend of useful teacher probability but mitigates degeneration. Next, we propose the repetition-aware pairwise alignment, which directly guides the model to generate probable alternative tokens.

### 4.1. FOCUS: FOcus on Confident Token Under Teacher Supervision

**FOCUS Training Objective.** Given a dataset $\mathcal{D} = \{s_i\}_{i=1}^{N}$, where each sequence is tokenized as $s_i = (s_{i,1}, \ldots, s_{i,T})$, let $z^T(s_{i,<t}) \in \mathbb{R}^V$ denote the teacher logits over the vocabulary $\mathcal{V}$ at position $t$. We define the student and teacher predictive distributions as

$$p_{i,t} \equiv p_\theta(\cdot \mid s_{i,<t}), \quad q_{i,t} \equiv \text{softmax}\left(\frac{z^T(s_{i,<t})}{\tau}\right),$$

We introduce a token-wise weight

$$w_{i,t}(s_{i,t}) = \left(q_{i,t}(s_{i,t})\right)^\beta + \left(1 - q_{i,t}(s_{i,t})\right)^\gamma, \quad \beta, \gamma \geq 0,$$

which emphasizes tokens where the teacher exhibits high confidence. Thus, naive KD is reformulated as

$$\mathcal{L}_{\text{FOCUS}} = \frac{1}{NT} \sum_{i=1}^{N} \sum_{t=1}^{T} \left(\tau^2 \sum_{v \in \mathcal{V}} w_{i,t}(v)\, q_{i,t}(v) \log \frac{q_{i,t}(v)}{p_{i,t}(v)}\right).$$

**Gradient Analysis of FOCUS** From the FOCUS objective

$$L_{\text{FOCUS}} = \sum_i w(q_i)\, q_i \log \frac{q_i}{p_i},$$

Applying the chain rule,

$$\frac{\partial L}{\partial a_k} = \sum_i \frac{\partial L}{\partial p_i} \frac{\partial p_i}{\partial a_k} = Z p_k - w_k q_k, \quad \text{where } Z = \sum_j w_j q_j.$$

The reweighted teacher distribution and gradient can be expressed as

$$\tilde{q}_k = \frac{w_k q_k}{Z}, \quad \nabla_a L_{\text{FOCUS}} = Z\,(p - \tilde{q}).$$

Thus, FOCUS preserves the standard KD form while effectively replacing the teacher distribution with a reweighted version $\tilde{q}$, making it easy to optimize. The detailed derivation is provided in Appendix B.

### 4.2. RePAIR: Repetition-aware PAIRwise Alignment

As discussed in Section 3, token-level guidance can alleviate repetition, but it cannot be applied at runtime since detecting the onset of a repetition loop requires access to future tokens. To address this, we propose RePAIR, a repetition-aware pairwise alignment that provides token-level corrective signals exactly where degeneration begins, guiding the student toward non-repetitive and contextually coherent generation. An example is provided in Appendix C.

#### 4.2.1. PAIRWISE DATA COLLECTION

Given a prefix length $k$, we generate model outputs $\hat{y}_i$ from inputs $s_{i,0:k}$. Using the Coverage metric described in Section 3, sequences with repetition above threshold are collected as negative samples $D_{\text{neg}}$. The index $r_i$ of the first repetition defines a shorter prefix $s_{i,0:(r_i-1)}$, from which we regenerate outputs to obtain positive samples $D_{\text{pos}}$. In practice, negative data are produced by the pruned model, while positive data are sampled from the unpruned model with top-$p$ decoding, yielding realistic pruned/unpruned comparisons. We set the threshold to 0.3.

#### 4.2.2. REPAIR TRAINING OBJECTIVE

We define a token-level pairwise margin loss to encourage the model to assign higher confidence to positive continuations than to negative ones.

**Training Objective.** Let $p_\theta(\cdot \mid \cdot)$ denote the model distribution. We compute token-level negative log-likelihood only for the continuation after the prefix $s_{i,0:(r_i-1)}^{\text{pre}}$.

$$\ell_i^+ = -\frac{1}{T_i^+ - r_i + 1} \sum_{t=r_i}^{T_i^+} \log p_\theta\big(s_{i,t}^{\text{pos}} \mid s_{i,0:(r_i-1)}^{\text{prefix}}, s_{i,<t}^{\text{pos}}\big),$$

$$\ell_i^- = -\frac{1}{T_i^- - r_i + 1} \sum_{t=r_i}^{T_i^-} \log p_\theta\big(s_{i,t}^{\text{neg}} \mid s_{i,0:(r_i-1)}^{\text{prefix}}, s_{i,<t}^{\text{neg}}\big).$$

We encourage the model to prefer the positive continuation over the negative one by a margin-ranking loss:

$$\mathcal{L} = \frac{1}{N} \sum_{i=1}^{N} \max\big(0,\ m + \ell_i^+ - \ell_i^-\big). \tag{12}$$

### 4.2.3. TOTAL LOSS FUNCTION

The final objective can be expressed as

$$L_{\text{total}} = L_{\text{CE}} + \alpha_1 \cdot L_{\text{FOCUS}} + \alpha_2 \cdot L_{\text{RePAIR}}$$

Unless otherwise specified, $\alpha_1$ is fixed at 0.05 and $\alpha_2$ at 1 for all experiments. A discussion on parameter selection is provided in Appendix F.

## 5. Experiments

### 5.1. Setup

We conduct experiments on the Llama family. We prune 25% of model width using LLMPruner and fine-tune on Alpaca, a standard benchmark for post-pruning knowledge recovery, using LoRA for two epochs. To stabilize the pruned model and maintain consistency with its pre-pruning behavior, we apply self-distillation. We further evaluate pruning rates of 35% and 45%, with results reported in Appendix I.

We evaluate our method on open-ended and instruction-based generation. Following prior work, for open-ended generation, we sample 1,000 instances from WikiText-103 and generate 100 tokens from a 50-token prefix. For instruction-based generation, we use 1,000 prompts from Self-Instruct (Wang et al., 2023). Across both tasks, we apply top-$p$ sampling ($p = 0.9$) and report both performance and degeneration metrics. We include both to ensure that methods designed to mitigate repetition do not excessively degrade task performance.

**Performance Metrics** We report performance using the following metrics:

- **Perplexity**: Perplexity (PPL) measures how well a language model predicts the next token, with lower values indicating better predictive performance.

- **BERTScore (BS):** BERTScore evaluates the semantic similarity between generated text and reference text by computing the cosine similarity of contextualized embeddings from a pretrained BERT model (Zhang et al., 2020). We report the F1 score.

- **Zero-shot Accuracy**: It reflects the model's ability to apply its general knowledge and reasoning skills. Details on the tasks and evaluation settings can be found in Appendix D.

- **MAUVE**: MAUVE measures the distributional similarity between generated and reference sentences in the embedding space (Pillutla et al., 2021).

**Degeneration Metrics** To capture the degeneration of models, we analyze:

- **Unique $n$-gram**: Unique n-gram rate is defined as $100 \times \big(1 - \frac{1}{N} \sum_{i=1}^{N} \frac{|\text{Unique } n\text{-gram}(\text{sentence}_i)|}{|\text{Total } n\text{-gram}(\text{sentence}_i)|}\big)$. We set $n = 3, 4, 5, 6$ to capture both word-level and phrase-level diversity (Xu et al., 2022).

- **Expected 1-gram Diversity (EAD$_1$):** EAD$_1$ quantifies lexical diversity by comparing the number of unique tokens in the generated text to the expected count under random sampling. A higher value indicates greater diversity, and it is formally defined as $\frac{N}{V_{\text{eff}}\left(1 - \left(1 - \frac{1}{V_{\text{eff}}}\right)^C\right)}$, where $N$ is the number of unique tokens in the generated text, $C$ is the total number of generated tokens, and $V_{\text{eff}}$ is the effective vocabulary size (Liu et al., 2022).

- **CREP**: CREP computes the fraction of sentences dominated by repeated $n$-grams (above 30%). For each sentence, we evaluate repetition across $n$-gram sizes $r \in [4, 16]$ and mark it as degenerated if any $r$ exceeds the threshold. This metric provides an intuitive measure of sentence-level repetition. Implementation details are provided in Appendix G.

**Comparison Methods.** We compare our method with four training-based baselines: (1) KD, which fine-tunes the pruned model via knowledge distillation; (2) UL-Token, which penalizes previously generated tokens (Welleck et al., 2019); (3) ScaleGrad (SG), which re-normalizes probabilities for non-generated tokens (Lin et al., 2021); (4) DITTO, which constructs sentence-repetition data and penalizes repetitive generations (Xu et al., 2022). Implementation details and hyperparameters are provided in Appendix E.

### 5.2. Open-ended Generation Task

As shown in Table 3, when applied alone, RePAIR outperforms most baselines across multiple metrics. For instance, it achieves the highest MAUVE score of 0.68 and the lowest CREP score of 2.23, indicating that it generates text more closely resembling real data with minimal degeneration.

*Table 3.* Results of open-ended generation on the WikiText-103 dataset. Best per block in **bold**, second best underlined. CREP and Unique $n$-gram are reported as percentage values.

| Method | PPL ($\downarrow$) | 0-shot ($\uparrow$) | MAUVE ($\uparrow$) | CREP ($\downarrow$) | Unique $n$-gram | | | |
|---|---|---|---|---|---|---|---|---|
| | | | | | $n=3$ | $n=4$ | $n=5$ | $n=6$ |
| **Llama-3.1-8B** | | | | | | | | |
| KD | 21.69 | 60.39 | 0.61 | 7.3 | 13.28 | 9.99 | 8.05 | 6.79 |
| + UL | **21.67** | 61.15 | 0.61 | 5.37 | 11.79 | 8.68 | 6.85 | 5.70 |
| + SG | 21.69 | 60.88 | 0.66 | 7.8 | 12.87 | 9.69 | 7.85 | 6.66 |
| + DITTO | 22.07 | 60.62 | 0.63 | 5.27 | 11.38 | 8.14 | 6.22 | 5.01 |
| + RePAIR | 22.05 | **61.36** | **0.68** | **2.23** | **8.36** | **5.36** | **3.74** | **2.78** |
| FOCUS | **22.32** | 60.03 | 0.64 | 1.73 | 6.22 | 3.82 | 2.62 | 1.94 |
| + UL | 23.20 | 60.45 | 0.69 | 0.77 | 4.30 | 2.43 | 1.51 | 1.07 |
| + SG | 22.71 | 60.47 | 0.70 | 0.87 | 4.86 | 2.89 | 1.96 | 1.47 |
| + DITTO | 22.88 | 60.54 | 0.72 | **0.5** | 4.49 | 2.42 | 1.46 | 1.11 |
| + RePAIR | 23.12 | **60.86** | **0.73** | 0.57 | **3.68** | **1.92** | **1.14** | **0.75** |
| **Llama-2-13B** | | | | | | | | |
| KD | **13.90** | 64.10 | 0.81 | 1.13 | 6.84 | 4.02 | 2.49 | 1.67 |
| + UL | **13.90** | 64.28 | 0.79 | 1.50 | 6.73 | 4.02 | 2.58 | 1.78 |
| + SG | **13.90** | 64.15 | **0.82** | 1.20 | 6.87 | 4.08 | 2.59 | 1.78 |
| + DITTO | 14.19 | 63.85 | 0.74 | 1.30 | 7.14 | 4.24 | 2.70 | 1.84 |
| + RePAIR | 13.95 | **64.36** | 0.81 | **0.50** | **5.55** | **2.97** | **1.67** | **1.01** |
| FOCUS | **15.41** | 63.69 | 0.84 | 0.13 | 2.84 | 1.20 | **0.55** | 0.29 |
| + UL | 15.74 | 63.57 | 0.85 | 0.07 | 2.40 | 0.94 | 0.38 | 0.17 |
| + SG | 15.47 | 63.77 | 0.84 | 0.10 | **2.57** | **1.07** | 0.47 | **0.23** |
| + DITTO | 15.85 | 63.52 | 0.86 | 0.03 | 2.46 | 0.93 | 0.37 | 0.16 |
| + RePAIR | 15.50 | **63.91** | **0.87** | **0.00** | 2.24 | 0.82 | 0.32 | 0.13 |
| Wiki-103 | - | - | - | - | 2.62 | 1.10 | 0.52 | 0.24 |

Moreover, it exhibits a distribution of unique $n$-gram scores comparable to that of the original WikiText-103 dataset. When combined with FOCUS, it consistently improves all metrics while incurring only a slight increase in perplexity. On average, the MAUVE score improves by about 0.06 across methods, accompanied by a marked decrease in the CREP metric, indicating a significant reduction in repetitive generations. Notably, RePAIR achieves both the highest MAUVE score and unique $n$-gram distributions that closely match real WikiText-103 text. Overall, these results demonstrate that our proposed methods substantially mitigate text degeneration while preserving perplexity and downstream task performance, indicating that the proposed training strategy incurs no meaningful performance degradation across both stand-alone and combined settings.

### 5.3. Instruction-based Generation Task

For instruction-based generation, since the output length for each instruction may vary across methods, we use $EAD_1$ as the primary diversity metric instead of unique $n$-gram diversity, as the latter does not account for differences in generation length. As illustrated in Table 4, without FOCUS, the model achieves the lowest degeneration performance despite recording the best perplexity among baselines. In contrast, RePAIR attains a lower CREP score of 0.63 and the high-

*Table 4.* Results of instruction-based generation on Self-Instruct dataset. Best per block in **bold**, second best underlined. CREP is reported as percentage values. (*BS: **BERTScore**)

| Method | PPL ($\downarrow$) | 0-shot ($\uparrow$) | CREP ($\downarrow$) | $EAD_1$ ($\uparrow$) | BS* ($\uparrow$) |
|---|---|---|---|---|---|
| **Llama-3.1-8B-Instruct** | | | | | |
| KD | 25.65 | 62.32 | 1.97 | 0.28 | **0.49** |
| + UL | **25.52** | 62.01 | 3.10 | 0.28 | 0.48 |
| + SG | 25.53 | 62.30 | 2.23 | 0.28 | **0.49** |
| + DITTO | 25.59 | 61.97 | 1.30 | 0.28 | **0.49** |
| + RePAIR | 25.86 | **62.61** | **0.63** | **0.32** | **0.49** |
| FOCUS | 26.54 | **62.85** | 0.73 | 0.30 | **0.50** |
| + UL | 26.89 | 62.07 | 0.73 | 0.29 | 0.49 |
| + SG | **26.27** | **62.85** | 0.53 | 0.30 | **0.50** |
| + DITTO | 26.36 | 62.12 | 0.30 | 0.29 | **0.50** |
| + RePAIR | 26.20 | 62.61 | **0.23** | 0.31 | **0.50** |

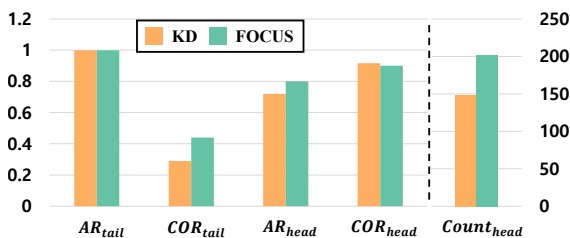

*Figure 2.* FOCUS analysis. Comparison of token distributions between naive KD and FOCUS.

*Table 6.* LLM-as-a-judge comparison results. * indicates FOCUS applied.

| Comparison | Win (A) | Tie | Win (B) |
|---|---|---|---|
| KD vs FOCUS | 40.10% | 19.80% | 40.10% |
| FOCUS vs RePAIR* | 36.00% | 22.60% | **41.40%** |
| UL* vs RePAIR* | 34.60% | 24.70% | **40.70%** |
| SG* vs RePAIR* | 35.50% | 24.30% | **40.20%** |
| DITTO* vs RePAIR* | 37.40% | 24.30% | **38.30%** |

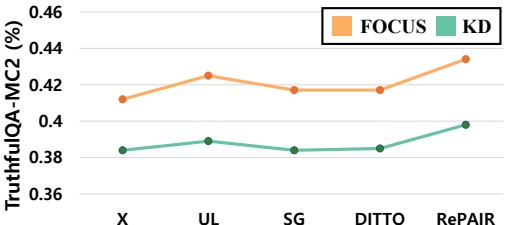

*Figure 3.* TruthfulQA-MC2 score. X indicates that no repetition-suppression technique was applied.

est $EAD_1$ score of 0.32, while maintaining a comparable perplexity to UL. This indicates that RePAIR produces the most diverse generations while preserving semantic fidelity, as reflected in BERTScore. When applied with FOCUS, all of the methods show consistent improvements across metrics with only a slight increase in perplexity. For example, it achieves lower CREP, higher $EAD_1$, and improved BERTScore, while maintaining stable zero-shot accuracy. This indicates that the method mitigates degeneration and enhances generation quality without incurring knowledge loss. Notably, when FOCUS is combined with RePAIR, most metrics achieve the best results among all methods. Overall, these findings demonstrate that our proposed approaches not only mitigate degeneration more effectively than existing baselines but also preserve general knowledge and semantic quality in instruction-following tasks.

*Table 5.* Repetition comparison of DPO and RePAIR under the WikiText-103 continuation setup using Llama-3.1-8B. For a clear comparison, KD and FOCUS are not applied here, and CE is used as the baseline.

| Method | PPL ($\downarrow$) | Unique $n$-gram | | | |
|---|---|---|---|---|---|
| | | $n=3$ | $n=4$ | $n=5$ | $n=6$ |
| CE | **23.05** | 12.88 | 9.68 | 7.77 | 6.56 |
| DPO | 23.29 | 9.76 | 6.53 | 4.64 | 3.45 |
| RePAIR | 23.72 | **8.28** | **5.53** | **3.97** | **3.01** |

### 5.4. LLM-as-a-Judge Evaluation

The current evaluation mainly relies on automatic metrics, which may not fully capture the perceptual quality of generated continuations. To provide a complementary assessment, we additionally conduct an LLM-as-a-judge evaluation. Following the experimental setup of Table 3, we generate Wiki continuation samples using Llama-3.1-8B and compare the outputs produced by different methods. We use GPT-120B (reasoning) as the judge. Detailed settings can be found in Appendix K.

As reported in Table 6, our method consistently outperforms the baselines in the LLM-as-a-judge evaluation, in addition to showing improvements on conventional automatic metrics. These results further support that the proposed method improves generation quality beyond what is captured by static repetition or perplexity-based measurements.

## 6. Analysis

**FOCUS Distribution Study** As discussed in Section 3, our method reweights token-level supervision to align the student with both high- and low-confidence regions of the teacher distribution, preserving teacher-preferred tokens while maintaining the relative structure in the tail. To empirically verify this behavior, we sample 200 responses generated by the teacher and compare the student's token probability distribution against the teacher's distribution. We quantify alignment using agreement rate (AR), $AR_{head} = \frac{|H_q \cap H_p|}{|H_q|}$ and $AR_{tail} = \frac{|T_q \cap T_p|}{|T_q|}$, and Pearson correlation (COR), where the head is defined by top-$p$ ($p = 0.9$). As shown in Figure 2, tail agreement remains stable while tail correlation increases substantially, indicating that our method preserves the relative probability geometry of the teacher's tail rather than merely matching the token set. This is consistent with our formulation in Section 3, which emphasizes structural alignment in low-confidence regions instead of indiscriminately flattening tail probabilities. In the head region, agreement increases substantially while the number of viable head candidates increases ($148.6 \rightarrow 202.1$). This suggests that our method maintains teacher-aligned high-probability tokens while redistributing probability mass among near-onset alternatives, expanding the set of plausible sampling candidates without sacrificing fidelity to the teacher.

**Robustness to Likelihood Instability** Based on Tables 3 and 4, we observe a consistent increase in PPL when applying our methods, which may reflect reduced likelihood stability or distributional drift. To examine whether this affects factual behavior, we evaluate LLaMA-3.1-8B on TruthfulQA. As shown in Fig. 3, although FOCUS increases PPL on WikiText, it improves TruthfulQA performance, and RePAIR achieves the largest overall gain. We attribute this to its token-level supervision, which explicitly guides the student toward teacher-preferred tokens. Similarly, FOCUS outperforms vanilla KD, consistent with prior work showing that capacity-aware distillation helps prioritize salient teacher signals under limited student capacity (Wu et al., 2024; Yao et al., 2025; Ko et al., 2024).

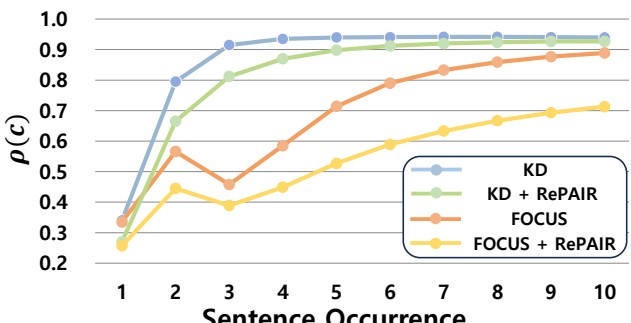

*Figure 4.* Persistence metric across methods.

**Onset-Level vs. Sequence-Level Supervision** RePAIR adopts a pairwise training paradigm similar to preference-based methods such as DPO (Rafailov et al., 2024), but differs in the granularity of its supervision. Whereas DPO provides sequence-level preferences, RePAIR operates at the onset-level by supplying corrective alternatives at repetition onset, directly targeting early loop entry. We compare RePAIR with DPO under the WikiText-103 continuation setting (Table 5). While DPO reduces repetition to some extent, RePAIR achieves larger gains, aligning with our analysis that fine-grained, onset-level corrective signals can be particularly effective for mitigating repetition. Implementation details are provided in Appendix J.

**Empirical Analysis of Loop Persistence.** To examine how FOCUS affects loop persistence, we conduct an empirical analysis using synthetic repetition contexts. We construct examples from WikiText-103 by taking 1,000 consecutive sentence pairs, using the first sentence as the prefix, and repeating the second sentence 10 times. We then measure $\rho(c) = \frac{a(c)}{a(c)+e(c)}$ as defined in Eq. (9), where $a(c)$ is the probability mass assigned to the next token on the repeated trajectory and $e(c)$ is the mass assigned to all other tokens. We report the average $\rho(c)$ over repeated positions.

Figure 4 shows that KD quickly saturates at a high $\rho(c)$ as sentence occurrence increases, indicating that the loop-continuing token increasingly dominates the escape mass and makes the loop harder to escape. In contrast, FOCUS delays this increase and saturation, suggesting reduced loop persistence. Combining FOCUS with RePAIR further suppresses $\rho(c)$, indicating an additional reduction in the tendency to remain in repetition loops.

**Comparison with Distribution-shaping Distillation.** FOCUS can be viewed as a distribution-shaping distillation objective, as it modifies the teacher distribution before applying distillation. To further examine whether other distribution-shaping distillation objectives can also mitigate repetition, we additionally compare FOCUS with ToDi, a token-level hybrid-KL distillation method that combines the

*Table 7.* Comparison with ToDi on Llama 3.1-8B, following the same experimental setting as Table 3. * indicates FOCUS applied.

| Method | MAUVE (↑) | CREP (↓) | Unique $n$-gram | | | |
|---|---|---|---|---|---|---|
| | | | $n=3$ | $n=4$ | $n=5$ | $n=6$ |
| ToDi | **0.73** | 6.73 | 12.13 | 9.00 | 7.19 | 6.05 |
| FOCUS | 0.64 | 1.73 | 6.22 | 3.82 | 2.62 | 1.94 |
| RePAIR* | **0.73** | **0.57** | **3.68** | **1.92** | **1.14** | **0.75** |

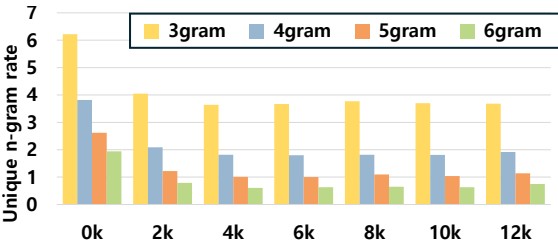

*Figure 5.* Number of pairwise data for RePAIR.

benefits of forward and reverse KL (Jung et al., 2025). As shown in Table 7, although ToDi improves generation quality, it is less effective than FOCUS in reducing repetition, as indicated by its higher unique $n$-gram repetition rates and CREP. Combined with RePAIR, FOCUS further reduces repetition while preserving generation quality, achieving the lowest repetition scores with a MAUVE score comparable to ToDi.

**Number of Pairwise Data** We collect 12k pairwise samples and analyze the amount of data required to mitigate text degeneration. As shown in Figure 5, only about 4k samples are sufficient to achieve repetition rates comparable to those obtained with the full dataset. This indicates that RePAIR is more data-efficient than DITTO, which consumes half of the training set for its auxiliary loss. As a result, our method preserves more data for standard training, enabling the model to learn broader knowledge.

## 7. Conclusion

We show that pruning increases repetition in language models and that token-level guidance can effectively mitigate this issue. To address repetition, we propose FOCUS, a token probability weighted distillation approach, and RePAIR, a pairwise margin-based objective that guides models toward better alternative tokens. Both methods consistently reduce repetition and improve generation quality.

## Impact Statement

This paper presents work whose goal is to advance the field of Machine Learning. There are many potential societal consequences of our work, none which we feel must be specifically highlighted here.

## Acknowledgements

This work was supported by the NRF grant funded by the Korea government (MSIT) (RS-2025-24803164), the Ministry of Trade, Industry Energy (MOTIE, Korea) under the project "An Unbreachable Multilayer AI-based Security Mechanism that Continuously Adapts and Evolves in Dynamic Conditions" (RS-20202653102), the Technology Innovation Program "Development of Navigation Technology Utilizing Visual Information Based on Vision-Language Models for Understanding Dynamic Environments in Non-Learned Spaces" (RS-2024-00445759).

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

## A. Repetition Analysis

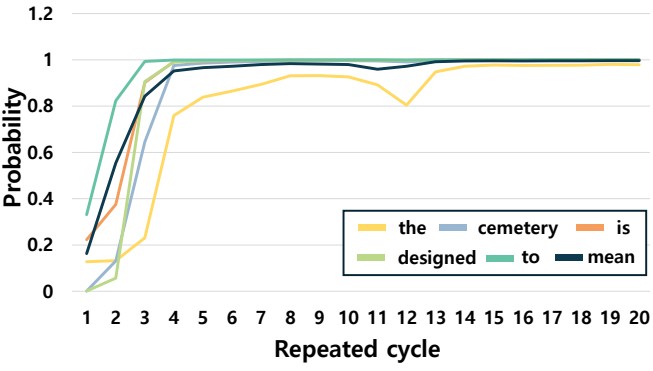

*Figure 6.* Token probabilities across repetition cycles.

Following prior work, we conduct a qualitative diagnostic experiment to illustrate how repetition loops amplify token probabilities over time, using the motivating example introduced in the Introduction. Specifically, we construct an input sequence consisting of a fixed prefix followed by repeated instances of a short sentence:

<prefix> ... for the deceased. The cemetery is designed to be a peaceful place. The cemetery is designed to be a peaceful place. The cemetery is designed to be a peaceful place. ...

At each step, we record the model's probability assigned to the observed next token and group these values by their within-sentence position. As illustrated in Figure 6, the resulting trajectories show that repeated-token probabilities rapidly saturate, indicating a near-deterministic repetition pattern once a loop is formed. While the persistence probability $\rho(c)$ in Eq. (6) is not directly observable, the saturation of mean probability over repeated tokens provides an empirical approximation of high persistence, suggesting that once the decoding trajectory enters a repetition loop, it becomes increasingly difficult for the model to escape.

## B. Gradient Analysis of FOCUS

In this section, we analyze why FOCUS has effects on the suppressing repetition loop.

**Gradient Analysis of Knowledge distillation.** The Knowledge Distillation (KD) loss is defined as the Kullback–Leibler (KL) divergence between the teacher distribution $q$ and the student distribution $p$:

$$L_{\text{KD}} = \sum_i q_i \log \frac{q_i}{p_i}. \tag{13}$$

Differentiating with respect to the softmax input $a$ (logit), we compute:

$$\frac{\partial L_{\text{KD}}}{\partial a_k} = -\sum_i q_i \frac{\partial \log p_i}{\partial a_k}.$$

The derivative of the log-softmax is:

$$\frac{\partial \log p_i}{\partial a_k} = \delta_{ik} - p_k,$$

where $\delta_{ik}$ is the Kronecker delta. Substituting back, we get:

$$\frac{\partial L_{\text{KD}}}{\partial a_k} = -q_k + \sum_i q_i p_k.$$

Since $\sum_i q_i = 1$, this simplifies to:

$$\frac{\partial L_{\text{KD}}}{\partial a_k} = p_k - q_k. \tag{14}$$

We can express it in vector form as:

$$\nabla_a L_{\text{KD}} = p - q. \tag{15}$$

**Gradient Analysis of FOCUS**    By modifying the KL objective, FOCUS defines the loss function as:

$$L_{\text{FOCUS}} = \sum_i w(q_i)\, q_i \log \frac{q_i}{p_i} \tag{16}$$

$$= \underbrace{\sum_i w(q_i)\, q_i \log q_i}_{\text{constant (independent of } a)} - \sum_i w(q_i)\, q_i \log p_i. \tag{17}$$

Differentiating $L_{\text{FOCUS}}$ with respect to $p_i$, we obtain:

$$\frac{\partial L}{\partial p_i} = -\frac{w_i q_i}{p_i}. \tag{18}$$

To compute the derivative with respect to the logits $a$, we need the softmax Jacobian:

$$\frac{\partial p_i}{\partial a_k} = p_i\,(\delta_{ik} - p_k), \tag{19}$$

where $\delta_{ik}$ is the Kronecker delta. By the chain rule, for each coordinate $k$:

$$\frac{\partial L}{\partial a_k} = \sum_i \frac{\partial L}{\partial p_i}\frac{\partial p_i}{\partial a_k} \tag{20}$$

$$= \sum_i \left(-\frac{w_i q_i}{p_i}\right) p_i\,(\delta_{ik} - p_k). \tag{21}$$

$$= \sum_i (-w_i q_i)\,(\delta_{ik} - p_k). \tag{22}$$

Next, we separate the summation into the cases $i = k$ and $i \neq k$:

$$\frac{\partial L}{\partial a_k} = (-w_k q_k)\,(1 - p_k) \;+\; \sum_{i \neq k} (-w_i q_i)\,(-p_k). \tag{23}$$

Finally, we obtain:

$$\frac{\partial L}{\partial a_k} = -w_k q_k \;+\; p_k \sum_i w_i q_i. \tag{24}$$

Let

$$Z := \sum_j w_j q_j > 0 \tag{25}$$

Then the gradient can be written as

$$\frac{\partial L}{\partial a_k} = Z p_k - w_k q_k. \tag{26}$$

Now, define a reweighted teacher distribution $\tilde{q}$ as

$$\tilde{q}_k \;\triangleq\; \frac{w_k q_k}{Z}. \tag{27}$$

Substituting, we obtain

$$\frac{\partial L}{\partial a_k} = Z\left(p_k - \tilde{q}_k\right). \tag{28}$$

In vector form, the gradient of FOCUS loss is:

$$\nabla_a L_{\text{FOCUS}} = Z\left(p - \tilde{q}\right), \quad \text{where } \tilde{q} = \frac{w(q) \odot q}{\sum_j w(q_j)q_j}, \tag{29}$$

To this end, FOCUS can be viewed as optimizing the student distribution with respect to a reweighted teacher distribution.

## C. An Example of Pairwise Data

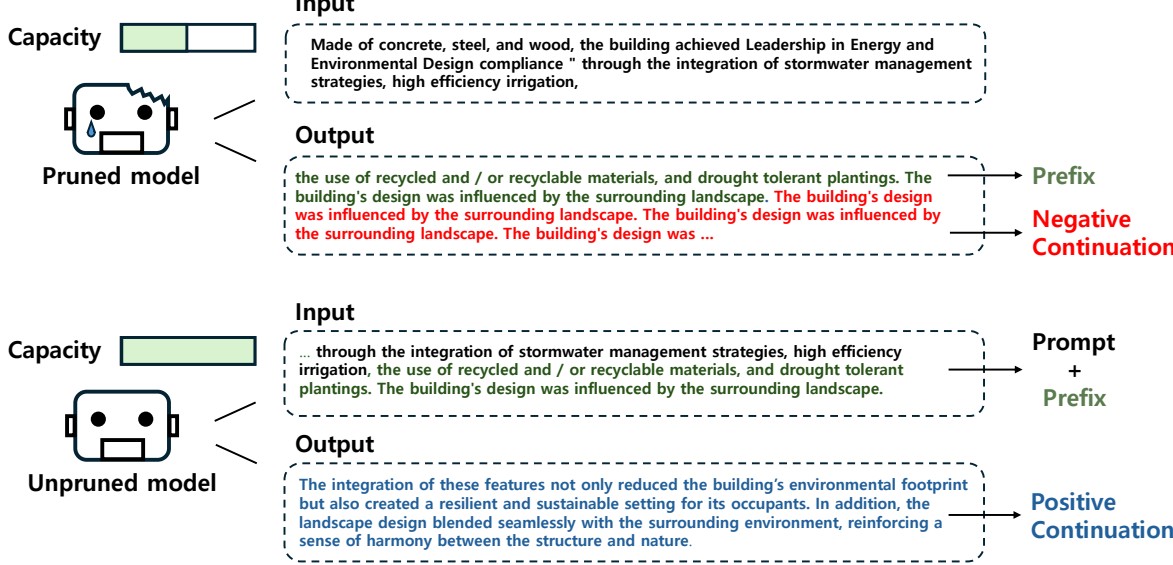

*Figure 7.* Example of pairwise data construction.

As illustrated in Figure 7, we first provide an input prompt to the pruned model and identify the onset of degeneration using the coverage metric. The sequence up to this point is designated as the prefix. We then feed the same input along with the prefix to the unpruned model to obtain a positive continuation that remains free of degeneration.

Finally, we construct training pairs of the form (prompt + prefix, negative continuation) and (prompt + prefix, positive continuation) to optimize the margin loss. In total, we collect 12k pairwise data. However, as shown in Figure 5, even 4k pairs are sufficient, highlighting the cost-effectiveness of the approach.

## D. Zero-shot Accuracy

**Evaluation Tasks**   We conduct zero-shot evaluations on the following benchmark tasks:

- **ARC_c**  (AI2 Reasoning Challenge): A multiple-choice science exam dataset that evaluates complex reasoning ability beyond simple fact recall.

- **PIQA**  (Physical Interaction Question Answering): A benchmark for testing physical commonsense reasoning, where models must choose the more plausible solution to everyday tasks.

- **BoolQ**  (Boolean Questions): A reading comprehension dataset consisting of yes/no questions with corresponding passages from Wikipedia.

*Table 8.* Zero-shot evaluation results on the Llama-3.1-8B model across six benchmarks. The best result within each block is highlighted in **bold**, while the second best is underlined.

| Model | Arc_c | PIQA | BoolQ | OpenQA | Hellaswag | Winogrande | Average |
|---|---|---|---|---|---|---|---|
| KD | 43.26 | 77.58 | 67.52 | 40.00 | 71.05 | 62.90 | 60.39 |
| + UL | **44.37** | 77.37 | 69.60 | **41.20** | 71.08 | 63.30 | 61.15 |
| + SG | 43.52 | 77.58 | 69.69 | 40.80 | 71.11 | 62.59 | 60.88 |
| + DITTO | 43.43 | 77.42 | 69.27 | 40.40 | 70.85 | 62.35 | 60.62 |
| + RePAIR | 43.17 | **77.86** | **70.55** | **41.20** | **71.86** | **63.54** | **61.36** |
| FOCUS | 42.92 | 77.26 | 65.87 | 40.20 | 70.70 | 63.22 | 60.03 |
| + UL | 43.34 | 77.15 | 67.55 | 40.80 | 70.42 | **63.46** | 60.45 |
| + SG | 42.92 | 77.20 | 68.17 | **41.00** | 70.69 | 62.83 | 60.47 |
| + DITTO | 42.66 | 77.15 | 69.08 | 40.80 | 70.46 | 63.06 | 60.54 |
| + RePAIR | **43.52** | **77.42** | **69.79** | 40.00 | **71.60** | 62.83 | **60.86** |

*Table 9.* Zero-shot evaluation results on the Llama2-13B model across six benchmarks. The best result within each block is highlighted in **bold**, while the second best is underlined.

| Model | Arc_c | PIQA | BoolQ | OpenQA | Hellaswag | Winogrande | Average |
|---|---|---|---|---|---|---|---|
| KD | 45.65 | 77.64 | 73.12 | **44.20** | 75.98 | 68.03 | 64.10 |
| + UL | **46.59** | **77.75** | **73.15** | 43.80 | 76.05 | **68.35** | 64.28 |
| + SG | 45.90 | 77.69 | **73.15** | 44.00 | 76.07 | 68.11 | 64.15 |
| + DITTO | 45.48 | 77.64 | 72.91 | 43.20 | 75.74 | 68.11 | 63.85 |
| + RePAIR | **46.59** | **77.75** | 72.78 | **44.20** | **76.80** | 68.03 | **64.36** |
| FOCUS | 45.99 | 77.26 | 73.15 | 43.20 | 74.98 | 67.56 | 63.69 |
| + UL | 45.90 | 77.31 | 72.97 | 43.00 | 74.68 | 67.56 | 63.57 |
| + SG | 45.82 | **77.48** | **73.33** | 43.20 | 75.00 | **67.80** | 63.77 |
| + DITTO | 45.65 | 77.26 | 72.72 | 43.20 | 74.66 | 67.64 | 63.52 |
| + RePAIR | **46.16** | 77.26 | 72.42 | **43.80** | **76.27** | 67.56 | **63.91** |

- **OpenQA** (Open-domain Question Answering): A task that requires answering factoid questions based on open-domain knowledge, without access to a fixed context passage.

- **Hellaswag**: A commonsense reasoning benchmark where the model must select the most plausible continuation of a given context.

- **Winogrande**: A large-scale pronoun resolution dataset designed to test commonsense reasoning through fill-in-the-blank style questions.

# E. Implementation Details

In this section, we provide the implementation details of baseline methods. All of the methods are implemented using huggingface framework and are based on the official implementation.

**Knowledge Distillation**  Knowledge distillation (KD) is a widely used technique to transfer knowledge from a large teacher model to a smaller student model.

$$L_{\text{KD}} = T^2 \cdot \text{KL}\left(\text{softmax}\left(\tfrac{z_t}{T}\right) \,\Big\|\, \text{softmax}\left(\tfrac{z_s}{T}\right)\right), \tag{30}$$

where $z_t$ and $z_s$ denote the logits of the teacher and student, respectively, and $T$ is the temperature parameter that controls the smoothness of the distributions. In our experiments, we adopt a temperature of $T = 2$, which is generally used and provides a good balance between stable training and effective knowledge transfer.

*Table 10.* Zero-shot evaluation results on the Llama-3.1-8B-Instruct model across six benchmarks. The best result within each block is highlighted in **bold**, while the second best is underlined.

| Model | Arc_c | PIQA | BoolQ | OpenQA | Hellaswag | Winogrande | Average |
|-------|-------|------|-------|--------|-----------|------------|---------|
| KD | 45.39 | 78.02 | 73.82 | 39.60 | 70.87 | 66.22 | 62.32 |
| + UL | 45.39 | 77.75 | 73.18 | 39.00 | 70.84 | 65.90 | 62.01 |
| + SG | **45.65** | 78.07 | 73.21 | 39.40 | **70.95** | 66.54 | 62.30 |
| + DITTO | 44.37 | 77.97 | 72.14 | **40.00** | 70.78 | 66.30 | 61.93 |
| + RePAIR | 45.48 | **78.13** | **73.67** | 39.80 | 71.71 | **66.85** | **62.61** |
| FOCUS | 45.39 | **78.62** | 73.85 | **42.20** | 71.00 | **66.06** | **62.85** |
| + UL | **46.08** | 77.58 | 72.05 | 40.80 | 70.95 | 64.96 | 62.07 |
| + SG | 45.82 | 77.70 | **74.04** | **42.20** | 71.05 | 66.30 | **62.85** |
| + DITTO | 44.76 | 77.86 | 71.35 | 41.60 | 70.98 | 66.14 | 62.12 |
| + RePAIR | 45.56 | 77.74 | 73.52 | 41.80 | **71.75** | 65.27 | 62.61 |

**Unlikelihood Training**   Unlikelihood training aims to reduce the probability of generating repeated tokens by penalizing candidates that already appear in the previous context. Although sentence-level variants have been explored in prior work with smaller models, applying them to large-scale models such as LLaMA is challenging due to memory constraints. Therefore, we adopt a token-level variant.

Specifically, at step $t$, we define the set of negative candidates as

$$C_{\text{prev-context}}^t = \{x_1, \ldots, x_{t-1}\} \setminus \{x_t\}. \tag{31}$$

To combine UL loss with maximum likelihood training, we adopt a token-level objective:

$$\mathcal{L}_{\text{UL-token}}^t(p_\theta(\cdot \mid x_{<t}), C^t) = -\alpha \cdot \sum_{c \in C^t} \log\left(1 - p_\theta(c \mid x_{<t})\right) \; - \; \log p_\theta(x_t \mid x_{<t}). \tag{32}$$

In our experiments, we set $\alpha = 0.5$, which provides a balanced trade-off between aggressively suppressing repetitions and preserving overall fluency and perplexity.

**ScaleGrad**   ScaleGrad is a repetition-penalization method that rescales the gradient on specific tokens which appeared previously in the context.

$$\tilde{p}_i = \begin{cases} \dfrac{\gamma \cdot p_i}{\sum_{j=1}^{|\mathcal{S}_{\text{novel}}|} \gamma \cdot p_j + \sum_{j=1}^{|\mathcal{V}'|} p_j}, & \text{if } i \in \mathcal{S}_{\text{novel}}, \\[4ex] \dfrac{p_i}{\sum_{j=1}^{|\mathcal{S}_{\text{novel}}|} \gamma \cdot p_j + \sum_{j=1}^{|\mathcal{V}'|} p_j}, & \text{otherwise.} \end{cases}$$

A smaller value of $\gamma$ more aggressively suppresses repetition, but at the cost of deteriorating perplexity. In the original paper, the authors experimented with $\gamma \in \{0.2, 0.5, 0.8\}$, and selected the value for each task accordingly. In our experiments, we set $\gamma = 0.5$, as it achieves a good balance between mitigating repetition and preserving perplexity.

**DITTO**   The authors focus on the self-reinforcement effect at the sentence level. To penalize repetition, they construct pseudo-repetition sentences and define the loss function as follows:

$$\mathcal{L}_{\text{DITTO}}^{n,l}(\mathcal{P}_\theta(x_{n,l} \mid \mathbf{x}_{<n,l})) = -\log\Big(1 - \big|\mathcal{P}_\theta(x_{n,l} \mid \mathbf{x}_{<n,l}) - \lambda \cdot \mathcal{P}_\theta^*(x_{n-1,l} \mid \mathbf{x}_{<n-1,l})\big|\Big).$$

Following the baseline setting, we use half of the training data for pseudo-repetition penalization. To construct pseudo-repetition data, the instruction and input are taken as a prefix, and the output is repeatedly appended until the maximum sequence length is reached. In accordance with the original implementation, we employ an MSE loss with $\lambda = 0.5$.

## F. Parameter Search

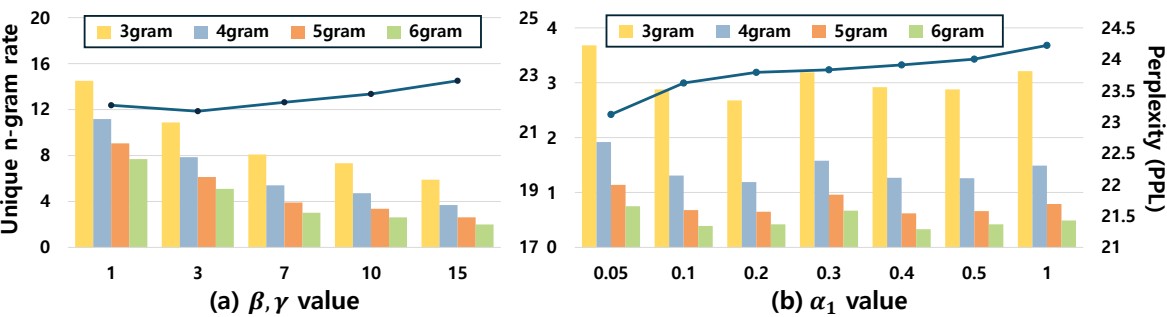

*Figure 8.* Parameter study of our methods on open-ended generation. (a) Unique $n$-gram with varying $\beta, \gamma$ values. (b) Unique $n$-gram with varying $\alpha_1$ values.

$\beta, \gamma$ **Selection**    We conduct an ablation study on the hyperparameters $\beta, \gamma$ of FOCUS. We generate outputs on the open-ended WikiText-103 generation task described above and evaluate them using the unique $n$-gram metric. As shown in Figure 8-(a), increasing these parameters consistently improves the overall unique $n$-gram, indicating that the model produces more diverse sentences. However, larger values slightly increase perplexity, since they encourage the model to rely more on high-confidence tokens during the distillation process. Nevertheless, as noted in Section 5.2, higher values lead to improvements in generation-related metrics, suggesting that perplexity alone should not be regarded as the sole criterion for evaluating model quality.

$\alpha_1$ **Selection**    As shown in Figure 8-(b), FOCUS influences perplexity (PPL) as $\alpha_1$ increases, making the choice of $\alpha_1$ crucial for controlling model perplexity. Following the main experiments, we fix $\beta$ at 15. To tune $\alpha_1$, we compute PPL and track the number of unique n-grams as its value increases. As shown in Figure 8, PPL deteriorates sharply when $\alpha_1$ is increased from 0.05 to 0.1, while the n-gram rate saturates from 0.1 onward. Therefore, we set $\alpha_1 = 0.05$ to balance PPL and $n$-gram repetition reduction.

# G. CREP: Coverage-based Repetition Metric

---

**Algorithm 1** CREP: Coverage-based Repetition Detection

---

**Input:** Dataset of generated texts $\mathcal{D}$, $n$-gram range $[r_{\min}, r_{\max}]$, global coverage threshold $\theta$
**Output:** CREP score in $[0, 1]$
$deg_{\text{count}} \leftarrow 0$
**for all** $y \in \mathcal{D}$ **do**
  $t \leftarrow \text{TOKENIZE}(y)$
  $best_{\text{cov}} \leftarrow 0$
  **for** $r \leftarrow r_{\min}$ **to** $r_{\max}$ **do**
    **if** $|t| < r + 1$ **then**
      **continue**
    **end if**
    $(g^{\star}, pos) \leftarrow \text{NGRAM\_WITH\_POSITIONS}(t, r)$
    **if** $|pos| < 2$ **then**
      **continue**
    **end if**
    $\Delta \leftarrow \{pos_{i+1} - pos_i \mid i = 1, \ldots, |pos| - 1\}$
    $d^{\star} \leftarrow \text{MODE}(\Delta)$
    Find smallest $i$ such that $pos_{i+1} - pos_i = d^{\star}$
    $s_1 \leftarrow pos_i$
    $s_2 \leftarrow s_1 + d^{\star}$
    $p \leftarrow t[s_1 : s_2]$           (candidate repeated segment)
    $u \leftarrow t[s_2 : |t|]$           (tail after first repetition)
    $cov_{\text{full}} \leftarrow \text{GLOBALCOVERAGE}(p, u, t)$
    $best_{\text{cov}} \leftarrow \max(best_{\text{cov}}, cov_{\text{full}})$
  **end for**
  **if** $best_{\text{cov}} \geq \theta$ **then**
    $deg_{\text{count}} \leftarrow deg_{\text{count}} + 1$
  **end if**
**end for**
**return** $CREP \leftarrow deg_{\text{count}} / |\mathcal{D}|$

---

To more rigorously detect text degeneration, we evaluate repetition across a broad range of $n$-gram lengths. For each generated sentence, we sweep $r$ from 4 to 16 and identify the most frequently recurring $r$-gram. We then reconstruct the full repeated segment and measure how much of the remaining output it covers globally. A sentence is marked as degenerate if its maximum coverage across all $r$ exceeds a predefined threshold. The full procedure is summarized in Algorithm 1.

## H. Experiment on Other Pruning Methods

*Table 11.* Results of open-ended generation on the WikiText-103 dataset for Llama 3.2-3B with Shortened-LLaMA after removing 7 blocks. Best per block in bold, second best underlined. CREP and Unique $n$-gram are reported as percentage values. * indicates FOCUS applied.

| Method | PPL ($\downarrow$) | 0-shot ($\uparrow$) | MAUVE ($\uparrow$) | CREP ($\downarrow$) | Unique $n$-gram | | | |
| --- | --- | --- | --- | --- | --- | --- | --- | --- |
| | | | | | $n=3$ | $n=4$ | $n=5$ | $n=6$ |
| KD | **32.73** | **52.43** | 0.55 | 10.57 | 16.03 | 12.56 | 10.46 | 9.07 |
| FOCUS | 33.05 | 51.92 | **0.76** | 2.20 | 6.60 | 4.01 | 2.73 | 2.03 |
| UL* | 33.18 | 51.93 | 0.74 | 2.30 | 6.75 | 4.17 | 2.93 | 2.24 |
| SG* | 33.18 | 51.99 | 0.70 | 1.53 | 5.67 | 3.37 | 2.26 | 1.66 |
| DITTO* | 34.10 | 52.18 | 0.70 | 1.00 | 5.09 | 2.67 | 1.58 | 1.04 |
| RePAIR* | 33.70 | 52.18 | **0.76** | **0.80** | **4.29** | **2.11** | **1.17** | **0.73** |

*Table 12.* Results of open-ended generation on the WikiText-103 dataset for Llama 3.2-3B with FLAP at pruning ratio 0.25. Best per block in bold, second best underlined. CREP and Unique $n$-gram are reported as percentage values. * indicates FOCUS applied.

| Method | PPL ($\downarrow$) | 0-shot ($\uparrow$) | MAUVE ($\uparrow$) | CREP ($\downarrow$) | Unique $n$-gram | | | |
| --- | --- | --- | --- | --- | --- | --- | --- | --- |
| | | | | | $n=3$ | $n=4$ | $n=5$ | $n=6$ |
| KD | 32.47 | 50.76 | 0.48 | 7.30 | 14.58 | 11.18 | 9.20 | 7.93 |
| FOCUS | **31.85** | **51.18** | 0.69 | 1.43 | 6.97 | 4.32 | 3.06 | 2.34 |
| UL* | 31.91 | 51.16 | **0.71** | 1.53 | 6.82 | 4.16 | 2.85 | 2.16 |
| SG* | 31.97 | 51.12 | 0.69 | 0.83 | 6.05 | 3.58 | 2.41 | 1.79 |
| DITTO* | 32.59 | 50.90 | 0.61 | 0.50 | 5.16 | 2.74 | 1.61 | 1.03 |
| RePAIR* | 32.59 | 50.89 | **0.71** | **0.37** | **4.39** | **2.25** | **1.34** | **0.91** |

*Table 13.* Results of open-ended generation on the WikiText-103 dataset for Qwen 2.5-3B with LLMPruner at pruning ratio 0.25. Best per block in bold, second best underlined. CREP and Unique $n$-gram are reported as percentage values. * indicates FOCUS applied.

| Method | PPL ($\downarrow$) | 0-shot ($\uparrow$) | MAUVE ($\uparrow$) | CREP ($\downarrow$) | Unique $n$-gram | | | |
| --- | --- | --- | --- | --- | --- | --- | --- | --- |
| | | | | | $n=3$ | $n=4$ | $n=5$ | $n=6$ |
| KD | **23.45** | 57.31 | 0.67 | 6.73 | 13.79 | 10.35 | 8.33 | 7.01 |
| FOCUS | 24.09 | 57.57 | 0.78 | 0.87 | 6.65 | 4.10 | 2.82 | 2.10 |
| UL* | 24.33 | 57.61 | 0.72 | 1.50 | 6.71 | 4.26 | 3.03 | 2.33 |
| SG* | 24.20 | 57.47 | **0.80** | 1.13 | 6.19 | 3.82 | 2.68 | 2.03 |
| DITTO* | 24.60 | **58.14** | 0.75 | **0.40** | 5.05 | 2.80 | 1.73 | 1.14 |
| RePAIR* | 24.49 | 57.63 | 0.76 | 0.43 | **4.50** | **2.44** | **1.53** | **1.05** |

We further evaluate the generality of our method by applying it to pruned models obtained from different pruning methods and model families. Tables 11, 12, and 13 show that our method consistently reduces repetition not only for LLMPruner, but also for other width- and depth-pruning methods. Moreover, the improvement on Qwen demonstrates that the effectiveness of our method is not limited to the LLaMA family. Overall, these findings indicate that our method is robust across different pruning strategies and model architectures.

*Table 14.* Results of open-ended generation on the WikiText-103 dataset for Llama 3.1-8B with LLMPruner at 35% and 45% pruning ratios. Best per block in bold, second best underlined. Unique $n$-gram is reported as percentage values. * indicates FOCUS applied.

| Method | PPL ($\downarrow$) | MAUVE ($\uparrow$) | Unique $n$-gram | | | |
|---|---|---|---|---|---|---|
| | | | $n = 3$ | $n = 4$ | $n = 5$ | $n = 6$ |
| **LLMPruner 35%** | | | | | | |
| FOCUS | **26.26** | 0.72 | 5.76 | 3.45 | 2.33 | 1.73 |
| + UL | 26.80 | 0.71 | 5.06 | 2.97 | 2.00 | 1.48 |
| + SG | 26.39 | **0.73** | 5.21 | 3.10 | 2.13 | 1.59 |
| + DITTO | 26.72 | 0.70 | 4.50 | 2.36 | 1.40 | 0.89 |
| + RePAIR | 27.12 | 0.70 | **3.22** | **1.46** | **0.75** | **0.43** |
| **LLMPruner 45%** | | | | | | |
| FOCUS | **33.27** | 0.44 | 6.87 | 4.46 | 3.26 | 2.55 |
| + UL | 33.94 | 0.34 | 5.55 | 3.42 | 2.37 | 1.79 |
| + SG | 33.36 | 0.44 | 6.18 | 4.05 | 2.97 | 2.36 |
| + DITTO | 34.12 | 0.35 | 4.46 | 2.52 | 1.60 | 1.09 |
| + RePAIR | 34.12 | **0.46** | **3.40** | **1.62** | **0.89** | **0.55** |

## I. Aggressive Pruning Settings

To demonstrate whether the proposed methods remain effective under more aggressive sparsity, we additionally evaluate models pruned at 35% and 45% using LLMPruner. Prior work has shown that pruning ratios above 30% typically induce noticeable degeneration (Jaiswal et al., 2024). Therefore, these settings allow us to assess the robustness of our approach in regimes where degradation is more prominent. As illustrated in Table 14, both FOCUS and RePAIR consistently deliver reliable improvements on repetition-related metrics, including n-gram rate, across all pruning levels. This consistent gain demonstrates the robustness of our methods, indicating that they effectively suppress degeneration patterns even as model sparsity increases. Meanwhile, the decline in MAUVE as pruning ratios increase reflects the expected loss of model capacity and distributional fidelity under aggressive compression, rather than a limitation of our repetition-mitigation strategy. Recovering semantic richness under such severe compression may require additional capacity-recovery or representation-restoration techniques beyond the scope of this work.

## J. DPO Training Details

To compare token-level corrective supervision with sequence-level preference learning, we implement Direct Preference Optimization (DPO) (Rafailov et al., 2024) as a baseline. We train DPO on the positive (non-repetitive) and negative (repetitive) sample pairs used for RePAIR, ensuring a fair comparison in terms of data and supervision.

Following the original DPO formulation (Rafailov et al., 2024), the optimization objective is defined as

$$\mathcal{L}_{\text{DPO}} = -\log \sigma \left( \beta_{dpo} \left( \Delta_{\text{student}} - \Delta_{\text{teacher}} \right) \right), \tag{33}$$

where

$$\Delta_{\text{student}} = \log \pi_{\text{student}}(y^+ \mid x) - \log \pi_{\text{student}}(y^- \mid x), \tag{34}$$

$$\Delta_{\text{teacher}} = \log \pi_{\text{teacher}}(y^+ \mid x) - \log \pi_{\text{teacher}}(y^- \mid x). \tag{35}$$

We set the DPO temperature parameter to $\beta_{dpo} = 0.1$, consistent with prior work. Both DPO and RePAIR are evaluated under the same WikiText-103 continuation protocol used in Table 3 and are not combined with KD or FOCUS. This allows us to isolate the effect of supervision granularity (sequence-level vs. token-level) when mitigating degeneration.

# K. LLM-as-a-Judge

For each comparison, the judge is given two anonymized candidate continuations, denoted as A and B, and is asked to choose among A, B, or TIE based on overall generation quality, with particular attention to fluency, coherence, and repetition. To reduce positional bias, we randomize the order of the two candidates. When the preference changes after swapping the candidate order, we conservatively count the result as TIE.

**System Prompt.**

```
You are an expert evaluator for Wikipedia-style continuation quality.
You compare two candidate continuations for the same underlying example
and decide which one is better.

Priorities, in order:
1. Less degeneration and repetition
2. Better encyclopedic style and coherence
3. Better local continuation quality and factual plausibility
4. Better sentence completion and fewer broken endings

Important rules:
- The two candidates correspond to the same example and must be compared directly.
- Heavily penalize repetitive loops, duplicated spans, and obvious degeneration.
- Prefer outputs that read like Wikipedia continuations:
  neutral, expository, and coherent.
- Base your judgment only on the provided candidates.
- Return only valid JSON with one field:
  {"winner":"A"} or {"winner":"B"} or {"winner":"tie"}.
- Do not include explanation.
```

**User Prompt Template.**

```
Compare candidate continuation A and candidate continuation B for example
index {index}.

They are alternative continuations for the same Wikipedia-like source
context.
Judge which continuation is better overall. Focus on repetition,
degeneration, coherence, encyclopedic tone, and whether the continuation
remains well-formed.

Candidate A:
{text_a}

Candidate B:
{text_b}

Output exactly one JSON object with a single field named winner.
```

