# OpenReview forum: "FOCUS & RePAIR: Mitigating Text Degeneration via Token-Level Guidance For Pruned Large Language Models"
_ICML.cc/2026/Conference — ICML 2026 spotlight_

### Official Review · Reviewer_ttAD · 2026-03-06

**Soundness:** 3
**Presentation:** 3
**Significance:** 3
**Originality:** 3
**Overall Recommendation:** 5
**Confidence:** 4

**Summary:**

This paper takes a deep dive into the issue of text degradation worsening after pruning LLMs, and proposes effective mitigation strategies. The authors offer a unique micro-level perspective, framing the decoding process as a dynamic system and breaking down repetitive degradation into two components: “loop entry risk” and “loop persistence.” Based on these theoretical findings, the paper introduces two token-level fine-tuning objectives: FOCUS and RePAIR.

**Compliance With Llm Reviewing Policy:**

Affirmed.

**Final Justification:**

A good paper, a nice and detailed rebuttal, I believe this work deserves an Accept.

**Key Questions For Authors:**

Please refer the **Weaknesses** section :-)

**Limitations:**

I think the authors could further emphasize that the effectiveness of FOCUS and RePAIR depends on the pruning ratio. When pruning is aggressive, the model’s capability boundaries may be breached, leading to performance drops. So in practical applications, it’s important to select the right pruning ratio based on the specific context.

**Strengths And Weaknesses:**

### Strengths
1. This work doesn’t just scratch the surface with improvements—it digs into the degradation dynamics under nucleus sampling. The paper clearly explains through derivation that reducing repetitiveness requires increasing the “escape mass” within the nucleus. This theoretical angle provides solid backing for the design of the methods that follow.

2. FOCUS and RePAIR tackle the problem from different angles. FOCUS addresses “tail leakage” during distribution matching to lower entry risk, while RePAIR precisely targets degradation onset to break loop persistence.


### Weaknesses
1. The experimental results show that while combining FOCUS and RePAIR significantly reduces CREP repetition rates, it also causes a slight increase in perplexity. Even though the authors use TruthfulQA scores to show this doesn’t lead to factual degradation, the underlying mechanism behind the perplexity rise (e.g. whether it’s due to artificial disruption of the natural probability distribution smoothness) isn’t explored in enough depth.

2. In the FOCUS, the weight function $ w_{i,t}(s_{i,t})=(q_{i,t}(s_{i,t}))^{\beta}+(1-q_{i,t}(s_{i,t}))^{\gamma} $ is introduced. The hyperparameters $ \beta $ and $ \gamma $ are designed based more on rule-of-thumb than theory. While the appendix includes parameter search analysis, there’s no direct mathematical explanation derived from prior theory.

---

> ### Author Rebuttal · Authors · 2026-03-29
>
> We thank the reviewer for their thoughtful feedback. We have addreesed the questions as follows. Also, for some experiments, please see the anonymous link for tables and figures (https://anonymous.4open.science/r/exp_figures-B63F)
>
> [W1]
>
> As pointed out, the TruthfulQA results primarily serve to alleviate concerns about potential side effects (e.g., factual degradation) associated with the increase in perplexity, rather than explaining the underlying cause of the PPL rise itself.
>
> The increase in PPL stems from the objective of FOCUS. Unlike standard KD, FOCUS does not aim to exactly match the teacher distribution; instead, it deliberately reshapes the distribution by assigning higher weight to tokens where the teacher is more confident, in order to suppress structures that lead to repetition. As analyzed in Appendix B, this induces a modified optimization landscape, where the model is no longer directly optimized for likelihood under the original data distribution, leading to a slight increase in PPL.
> Based on this discussion, we will revise the paper to more clearly disentangle these discussions.
>
> [W2]
>
> We analyze repetition through the relative dominance of a repetition-continuation token $r$ compared to its alternatives. Rather than treating repetition as a strict condition, we view it as a probabilistic tendency: when the probability of $r$, denoted $q_r$, is large relative to competing tokens, the tendency to continue a repetitive pattern increases.
>
> Under FOCUS, the effective target distribution is defined as a reweighted teacher distribution $\tilde{q}_v = \frac{q_v w(q_v)}{\sum_j q_j w(q_j)}$, where $w(q) = q^k + (1 - q)^k$ and $k > 1$.
>
> To quantify relative dominance, we consider the ratio between the repetition token and the total mass of competing tokens: $\frac{\tilde q_r}{\sum_{v\neq r} \tilde q_v} = \frac{q_r w(q_r)}{\sum_{v\neq r} q_v w(q_v)}$
>
> We consider a regime where the repetition token is the most probable token but not overly dominant, i.e., $q_v < q_r$ and $q_r < 1/2$, i.e., $q_v < q_r < \frac{1}{2}$.
>
> For $k > 1$, the weighting function is $w(q) = q^k + (1 - q)^k$ with derivative $w'(q) = k \left(q^{k-1} - (1 - q)^{k-1}\right)$. For $q < \frac{1}{2}$, we have $w'(q) < 0$, and thus $q_r > q_v \Rightarrow w(q_r) < w(q_v)$.
>
> Applying this inequality term-wise, for all $v ≠ r$ we have $w(q_v) > w(q_r)$. Since $q_v ≥ 0$, we can multiply both sides by $q_v$ and sum over $v ≠ r$, which yields $\sum_{v \neq r} q_v w(q_v) > w(q_r) \sum_{v \neq r} q_v$.
>
> Using this inequality, we obtain
> $$\frac{q_r w(q_r)}{\sum_{v\neq r} q_v w(q_v)} < \frac{q_r w(q_r)}{w(q_r)\sum_{v\neq r} q_v} = \frac{q_r}{\sum_{v\neq r} q_v}$$
>
> Thus, FOCUS reduces the relative dominance of the repetition-continuation token and redistributes probability mass toward competing alternatives.
>
> Moreover, this suppression becomes stronger as $k$ increases. For $q < \frac{1}{2}$, the term $(1 - q)^k$ dominates $q^k$ as $k$ increases. Hence, $\frac{w(q_r)}{w(q_v)} \approx \left( \frac{1 - q_r}{1 - q_v} \right)^k$.
>
> Since $q_r > q_v$, we have $1 - q_r < 1 - q_v$, and thus the ratio $w(q_r) / w(q_v)$ decreases exponentially with $k$ in this regime. As the dominance is given by $q_r w(q_r) / \sum_{v\neq r} q_v w(q_v)$, this further increases the relative contribution of competing tokens in the denominator, leading to a stronger reduction in the relative dominance of the repetition-continuation token. For related experimental results, please see Reviewer oM7v [W2, Q1].
>
> (When $q_r > \frac{1}{2}$, the repetition token is strongly favored by the teacher distribution. In this regime, FOCUS would preserve this preference rather than explicitly suppressing it, as knowledge distillation relies on the assumption that the teacher distribution is reliable.)
>
> [Limitations]
>
> We agree with the reviewer that the effectiveness of our method depends on the pruning ratio. Under aggressive pruning, the model’s representational capacity can be severely degraded, making it difficult to disentangle repetition-specific effects from overall quality degradation.
>
> For this reason, we focus on moderate pruning regimes (25% in our main experiment), where the model retains sufficient capability and repetition can be meaningfully analyzed. As shown in our results (e.g., Appendix Table 9), higher pruning ratios lead to a rapid degradation in generation quality, which can confound the evaluation of repetition.
>
> We will clarify this point and its practical implications in the revised version.

---

> > ### Author Rebuttal · Reviewer_ttAD · 2026-04-01
> >
> > A good paper, a nice and detailed rebuttal, I believe this work deserves an Accept :-)

---

### Official Review · Reviewer_oM7v · 2026-03-11

**Soundness:** 3
**Presentation:** 3
**Significance:** 3
**Originality:** 3
**Overall Recommendation:** 5
**Confidence:** 3

**Summary:**

This paper investigates text degeneration in pruned LLMs, focusing on repetition loops that often arise during generation. The authors analyze degeneration from a token-level perspective by modeling decoding as a stochastic process over contexts. Within this framework, they decompose degeneration into two factors: loop entry risk, the probability of entering a repetition-prone region, and loop persistence, the probability of remaining within that region once entered. The analysis emphasizes how probability allocation within the nucleus sampling set influences repetition, suggesting that degeneration occurs when loop-continuing tokens dominate the sampling candidates while alternative tokens that could escape the loop receive insufficient probability.

Based on this analysis, the paper proposes two training objectives for post-pruning fine-tuning. FOCUS modifies knowledge distillation by emphasizing high-confidence teacher predictions to reduce probability leakage into teacher-suppressed tokens, while RePAIR introduces a pairwise objective using positive and negative continuations around detected repetition onsets to encourage non-repetitive alternatives. Experiments on open-ended text continuation and instruction-following generation show that these methods reduce repetition and improve generation quality, with the combination of FOCUS and RePAIR achieving the best results across several metrics.

**Compliance With Llm Reviewing Policy:**

Affirmed.

**Final Justification:**

The article aims to analyze the key question of how pruning amplifies repetition-loop degeneration, proposing an intuitive entry-persistence decomposition and two practical training-time interventions. The rebuttal substantively addressed my main concerns: direct measurement of leakage mass over T_ε(c) on natural contexts (W1/Q2), onset-position analysis supporting functional separation (W2/W3), and additional experiments across new models, pruning methods, and LLM-as-judge evaluation. The remaining issue is that the theoretical attribution is imprecise—FOCUS also reduces persistence-related quantities, weakening the claimed one-to-one mapping between methods and mechanisms—and the scope is limited to repetition-based degeneration. I raise my assessment from weak accept to accept, provided that the final version tempers the functional decomposition claims and acknowledges these scope limitations.

**Key Questions For Authors:**

1. **Empirical validation of the entry–persistence decomposition.**
   The paper proposes a conceptual decomposition of degeneration into loop entry risk and loop persistence, which motivates the design of FOCUS and RePAIR. However, the experiments do not directly measure these quantities. Could the authors provide empirical analyses that estimate loop entry frequency and loop persistence (e.g., the probability of remaining in a detected loop) for the baseline and the proposed methods? Such analysis would help verify whether FOCUS primarily reduces entry risk while RePAIR reduces persistence, as suggested by the theoretical framework.


2. **Evidence for the probability leakage mechanism.**
   The paper argues that pruning combined with naive knowledge distillation leads to probability leakage into teacher-suppressed tokens, which increases loop entry risk. However, the experiments do not directly measure the leakage quantity defined in the paper. Could the authors provide empirical measurements of this leakage mass (e.g., probability assigned to teacher-suppressed tokens) before pruning, after pruning with naive distillation, and after applying FOCUS?

3. **Sensitivity of results to the degeneration threshold τ.**
   The coverage threshold τ = 0.3 is used both to define degeneration in the CREP metric and to construct pairwise training data for RePAIR. Have the authors evaluated how sensitive the results are to different threshold choices (e.g., τ ∈ {0.1, 0.2, 0.4, 0.5})?

**Limitations:**

The paper does not explicitly discuss limitations or broader societal impacts. It would be helpful for the authors to briefly acknowledge that the current study focuses on repetition-based degeneration and may not directly address other generation pathologies. A short discussion of the scope and limitations of the proposed framework would improve completeness.

**Strengths And Weaknesses:**

**Strengths:**

1. **Addresses a practically relevant problem in LLM deployment.**
   The paper studies text degeneration in pruned large language models, which is an important issue for real-world LLM deployment where model compression is often necessary to meet computational constraints. Repetition loops are a common failure mode in compressed models, and improving generation quality under pruning is therefore practically valuable. The work focuses on post-pruning fine-tuning, a realistic setting in which improving generation behavior without increasing model size is particularly useful.

2. **Provides a clear conceptual perspective on repetition loops.**
   The paper proposes a conceptual decomposition of degeneration into loop entry risk and loop persistence, linking repetition behavior to the distribution of probability mass among candidate tokens during nucleus sampling. This perspective offers an intuitive way to reason about how token probabilities influence repetition dynamics and helps motivate the proposed training objectives.

3. **Empirical results demonstrate consistent improvements over several baselines.**
   The proposed methods are evaluated on both open-ended text continuation and instruction-following generation, and the experiments report multiple metrics capturing repetition, diversity, and semantic similarity. The results consistently show reductions in repetition and improvements in several generation quality metrics compared with multiple training-based baselines, suggesting that the approach is empirically effective.

4. **Methods are simple and practical to integrate into existing training pipelines.**
   The proposed objectives, FOCUS and RePAIR, are lightweight extensions of existing training paradigms such as knowledge distillation and pairwise preference training. Because they operate during post-pruning fine-tuning and do not require changes to inference-time decoding, they could be incorporated into existing compression workflows with relatively low implementation complexity.

**Weaknesses:**

1. **Incomplete justification of the probability leakage argument.**
The paper argues that pruning combined with naive knowledge distillation leads to probability leakage into teacher-suppressed tokens due to the use of forward KL divergence. While it is true that forward KL places little penalty on tokens with near-zero teacher probability, this property alone does not imply that the student distribution will assign higher probability to these tokens. The argument implicitly relies on capacity limitations introduced by pruning, but this interaction is not analyzed or empirically validated. Providing direct measurements of the proposed leakage mass or demonstrating its increase after pruning would strengthen the claim.


2. **Limited empirical validation of the theoretical decomposition.**
The paper proposes a theoretical decomposition of degeneration probability into loop entry risk and loop persistence. However, the experiments do not directly measure these quantities or validate the decomposition empirically. For example, the paper does not report metrics such as empirical loop entry frequency or persistence probabilities. As a result, it is unclear whether the improvements from FOCUS and RePAIR correspond to the mechanisms suggested by the theoretical analysis.

3. **Training objectives are not empirically disentangled.**
The paper conceptually attributes FOCUS to reducing loop entry risk and RePAIR to reducing loop persistence. However, both objectives modify the token probability distribution globally during training, and the experiments do not verify that each method primarily affects the component it is claimed to target. Additional analysis measuring entry rates or persistence within detected loops would strengthen the connection between the theoretical framework and the empirical results.

4. **Sensitivity to the coverage threshold τ is not analyzed.**
The degeneration metric relies on a fixed coverage threshold τ = 0.3, which is also used when constructing pairwise training data for RePAIR. Because this threshold affects both evaluation and training data generation, the reported improvements may depend on this design choice. A sensitivity analysis across different thresholds would strengthen the robustness of the results.

5. **Missing comparisons with decoding-based mitigation strategies.**
The evaluation focuses on training-based baselines but does not compare against commonly used decoding-time mitigation methods such as repetition penalties, no-repeat n-gram constraints, or alternative sampling strategies. Including such comparisons would help clarify the practical advantages of the proposed approach.


6. **Scope of degeneration studied is not clearly articulated.**
The paper focuses on repetition loops as the primary degeneration phenomenon. Prior work has discussed other generation pathologies in neural text generation, such as unnatural high-probability text and low-diversity outputs induced by likelihood-maximizing decoding [1,2]. While focusing on repetition is reasonable, the paper would benefit from clearly stating that the scope of the study is limited to repetition-based degeneration and discussing whether the proposed framework could extend to these other degeneration behaviors.

---

**References**

[1] Holtzman, A., Buys, J., Du, L., Forbes, M., & Choi, Y. (2020). *The Curious Case of Neural Text Degeneration.* ICLR.

[2] Vijayakumar, A., Cogswell, M., Selvaraju, R., Sun, Q., Lee, S., Crandall, D., & Batra, D. (2016). *Diverse Beam Search: Decoding Diverse Solutions from Neural Sequence Models.* AAAI.

---

> ### Author Rebuttal · Authors · 2026-03-29
>
> We sincerely thank the reviewer for their detailed and insightful feedback, which helped us improve the clarity and quality of our paper. We address the questions as follows. please see the anonymous link for tables and figures (https://anonymous.4open.science/r/exp_figures-B63F)
>
> [W2, Q1]
>
> To better present how the proposed design mitigates the loop persistence issue as supported by our theoretical analysis, we conducted additional validation experiments.
>
> Specifically, we construct synthetic repetition scenarios using WikiText. We extract pairs of consecutive sentences, use the first sentence as the context, and repeat the second sentence 20 times to form a repetitive continuation. On these sequences, we compute two quantities. First, we measure a proxy for repetition probability using the geometric mean of token probabilities along the repetition path: $\exp\left(\frac{1}{|\mathcal{A}(c)|} \sum_{t \in \mathcal{A}(c)} \log p(t \mid c)\right)$, where $\mathcal{A}(c)$ denotes the set of tokens that follow the repetition pattern. Second, we directly compute $\rho(c)$ as defined in Eq. (9) of Section 3.3. Specifically, we decompose the probability mass within the repeated sentence into continuation and escape components: tokens along the repetition path contribute to $a(c)$, while all remaining tokens contribute to $e(c)$, and compute $\rho(c) = \frac{a(c)}{a(c) + e(c)}$.
>
> As shown in Figure 1 and 2 in the link, from a persistence perspective, compared to KD, our method consistently yields lower overall probabilities and saturates at lower probability levels as repetition progresses. This indicates that the likelihood of sustaining repetition decreases, reducing the probability of entering and maintaining repetition loops.
>
> [W1, Q2, W3]
>
> We further analyze repetition from a leakage perspective, where the key factor is how much probability mass spreads toward repetition-inducing tokens.
>
> To quantify this, we compute a teacher win rate at the onset region(i.e., the first repetition step, excluding the original occurrence of the token) in the synthetic sentences constructed above. For each sample $i$, we compare the probability assigned to the repetition-continuation token $r_i$ by the teacher and the pruned-and-tuned model. The teacher win rate is defined as:
>
> $\text{WinRate} = \frac{1}{N} \sum_{i=1}^{N} \mathrm{I}\left(q_{\mathrm{teacher}}(r_i) > q_{\mathrm{model}}(r_i)\right)$
>
> where $\mathrm{I}(\cdot)$ is an indicator function.
>
> A lower teacher win rate implies that the model assigns higher probability to the repeated token than the teacher, which can be interpreted as increased leakage of probability mass toward repetition-inducing tokens.
>
> As shown in Figure 3 of the link, we compute this metric around the onset region using the same synthetic data described above. The results show that KD tends to have a lower teacher win rate compared to FOCUS. This suggests that KD allocates more leakage to repetition tokens at onset, increasing the risk of entering repetition loops.
>
> Overall, due to the inherently dynamic nature of the probabilistic generation process, it is difficult to attribute each method’s effect to a single factor in isolation. Nevertheless, the experiments (Figures 1, 2, and 3) indicate that both methods effectively suppress repetition loop components, including entry and persistence. We will revise the text in a future version to better clarify this point.
>
> [W4, Q3]
>
> We set τ = 0.3 in a somewhat conservative and empirical manner. In our setup with 100 generated tokens, this choice roughly corresponds to cases where a short phrase or sentence is repeated 2 times, allowing us to capture noticeable repetition while avoiding overly sensitive detection of minor overlaps.
> Since same τ is used both for constructing training data and for evaluation, there is a potential concern that the improvements may depend on this specific choice. To address this, we evaluate multiple τ ∈ {0.1, 0.2, 0.3, 0.4, 0.5}.
>
> As shown in Figure 7 in the linked materials, Across all  τ, our method consistently outperforms the baselines. At the same time, very small τ values tend to classify even trivial repetitions as repetition, while very large τ values fail to capture most repetition cases. Therefore, selecting an appropriate τ is important for ensuring the validity and discriminative power of the metric.
>
> [W5]
>
> We conducted additional experiments on pruned models with decoding strategies such as beam search and repetition penalty. Due to the limited space, please refer Reviewer nJHu [Additional] part.
>
> [W6, Limitations]
>
> As the reviewer pointed out, our work directly addresses degeneration caused by repetition and therefore does not explicitly handle other forms of degeneration. In fact, as shown in Table 9, while our method reduces repetition, it does not resolve the degradation in overall text quality, as reflected by the MAUVE score. We will add a section to clearly discuss the scope and limitations of our approach.

---

> > ### Author Rebuttal · Reviewer_oM7v · 2026-04-04
> >
> > We thank the authors for the rebuttal and the additional analyses. The threshold sensitivity study (W4) and the acknowledgment of scope limitations (W6) are adequately addressed. I have follow-up questions on the remaining concerns.
> >
> > **W1 & Q2: Probability Leakage Measurement**
> >
> > The teacher win rate metric is a useful addition and provides some evidence that KD assigns higher probability to repetition-continuation tokens at onset compared to FOCUS. However, this metric is computed only at repetition-specific tokens within synthetically constructed repetition sequences, which captures a narrow aspect of the leakage phenomenon described in the paper.
> >
> > The theoretical framework defines the teacher-suppressed set T_ε(c) in Eq. (11) and the student's leakage mass Δ_ε(c) immediately following Eq. (11) as the total probability the student assigns to all tokens in T_ε(c)—that is, all tokens where the teacher assigns near-zero probability, not only those that happen to continue a repetition loop. The teacher win rate, by contrast, compares probabilities at a single token (the repetition-continuation token) at onset positions in synthetic sequences. These are related but distinct quantities.
> >
> > Could the authors directly measure Δ_ε(c) as defined in the paper? Specifically, for a set of naturally occurring contexts from WikiText-103, compute the total probability mass assigned to tokens in T_ε(c) (defined relative to the unpruned teacher), and compare this quantity across three conditions: (1) the pruned model before fine-tuning, (2) the pruned model fine-tuned with KD, and (3) the pruned model fine-tuned with FOCUS. This would provide direct empirical validation of the central claim that pruning and naive distillation increase leakage into teacher-suppressed regions and that FOCUS reduces it.
> >
> > **W2 & W3: Disentangling FOCUS and RePAIR**
> >
> > I appreciate the authors' candid acknowledgment that "it is difficult to attribute each method's effect to a single factor in isolation." The paper explicitly frames FOCUS as targeting loop entry risk through leakage suppression (Section 3.4, "Tail leakage increases loop entry risk") and RePAIR as targeting loop persistence by promoting escape alternatives at loop-sensitive contexts (Section 3.3, "Nucleus alternatives and escape mass"). This functional decomposition is central to how the two contributions are motivated and presented. However, the rebuttal's analysis does not validate this separation.
> >
> > Specifically, Figures 1 and 2 show that FOCUS alone (green) already substantially reduces persistence-related quantities (ρ and geometric mean token probability) compared to KD (blue), indicating that FOCUS affects both entry risk and persistence rather than entry risk alone. Meanwhile, Figure 3 (teacher win rate) only compares KD against FOCUS, without including RePAIR or KD+RePAIR conditions, leaving RePAIR's independent effect on entry risk entirely unexamined. While Figures 1 and 2 do confirm that RePAIR reduces persistence (orange below blue; red below green), the claimed one-to-one mapping—FOCUS targets entry, RePAIR targets persistence—is not supported by the evidence presented. Could the authors provide additional analysis that isolates each method's effect on entry and persistence separately?

---

> > > ### Author Response · Authors · 2026-04-07
> > >
> > > * We have updated annoymous link for tables and figures.
> > >
> > > **[W1 & Q2: Probability Leakage Measurement]**
> > >
> > > We conducted the experiment requested by the reviewer. For naturally occurring contexts from WikiText-103, we construct the teacher-suppressed set 𝑇𝜖(𝑐) based on the teacher distribution. We set 𝜖=0.1 and 0.01, as this threshold captures tokens that the teacher assigns low probability to and would rarely sample in practice. To reflect the impact of the actual sampling process $S_p(c)$ discussed in our original paper, we quantify the leakage mass over the 𝑇𝜖(𝑐) vocabulary not also sampled by each student model under nucleus (top-p, 0.9) decoding. As shown in Figure 4 in the supplementary link, as compared to KD, FOCUS exhibits lower leakage for the mass region that the teacher suppresses and the student would not sample. This phenomenum is consistently observed for both 𝜖= 0.1 and 0.01.
> > >
> > > Following the reviewer’s request, we also include the pruned model before fine-tuning. This model serves as a reference point to illustrate how pruning alone redistributes probability mass. We observe that naive KD tends to increase leakage relative to this baseline, while FOCUS counteracts this effect and better preserves the teacher-aligned suppression of low-probability tokens.
> > >
> > >
> > > [W2 & W3: Disentangling FOCUS and RePAIR]
> > >
> > > As the reviewer pointed out, Figures 1 and 2 in the supplementary link primarily analyze persistence-related behavior. As requested, we have also updated Figure 3, which now includes the KD+RePAIR and FOCUS+RePAIR results.
> > >
> > > We also conducted an additional experiment to more clearly isolate the impact on repetition entry behavior in a setting that reflects real-world scenarios. Specifically, we use the Wiki continuation data described in Section 5.2 of our original paper and measure when the onset (i.e., repetition entry index) occurs during generation. As shown in Table 9 in the link, there is no significant difference in the entry position between KD and KD+RePAIR. In contrast, applying FOCUS highly delays the entry point, i.e., the model at least keeps generation for a longer span before repetition begins. This provides empirical evidence that FOCUS shifts the onset of repetition to later positions.

---

### Official Review · Reviewer_nJHu · 2026-03-12

**Soundness:** 2
**Presentation:** 3
**Significance:** 3
**Originality:** 2
**Overall Recommendation:** 4
**Confidence:** 4

**Summary:**

This paper studies text degeneration in pruned LLMs and analyzes it from a token-level decoding dynamics perspective. It shows that pruning can increase the risk of entering and persisting in repetition loops even when perplexity remains similar. To address this issue, This paper proposes two training objectives: FOCUS, a weighted distillation method that emphasizes high-confidence teacher tokens, and RePAIR, a pairwise alignment approach that encourages alternative continuations at repetition onsets. Experiments on open-ended and instruction-based generation tasks show that these methods reduce repetition and improve generation quality in pruned LLMs.

**Compliance With Llm Reviewing Policy:**

Affirmed.

**Final Justification:**

Fully resolved - My concerns have been adequately addressed.

**Key Questions For Authors:**

See weakness.

[Additional]:

Could the authors discuss whether similar degeneration mitigation could be achieved without additional retraining, for example through decoding-time interventions or lightweight adaptation methods?

**Limitations:**

yes

**Strengths And Weaknesses:**

**Strengths**

1. The paper studies an important practical issue in LLM compression: pruning can increase text degeneration (e.g., repetition loops) even when standard metrics remain stable.
2. The proposed methods are relatively simple and can be integrated into existing post-pruning training pipelines.
3. The experiments evaluate both open-ended and instruction-based generation and report several generation-quality and repetition metrics.

**Weaknesses**

1. **Missing important baselines.** The comparison excludes several relevant distillation objectives (e.g., reverse-KL–based or hybrid KL formulations such as MiniLLM [1-3]), which are directly related to token-level probability alignment and could provide stronger baselines.

2. **Limited experimental coverage.** The experiments are restricted mainly to pruned LLaMA-family models and a small set of generation datasets. The evaluation does not include a broader range of model families (e.g. Qwen-family) or more advanced reasoning benchmarks (e.g., math or multi-step reasoning tasks), making it unclear whether the method generalizes beyond the tested setting.

3. **Insufficient evaluation methodology.** The evaluation relies primarily on automatic metrics (e.g., PPL, MAUVE, CREP) and does not include more recent evaluation practices such as LLM-as-a-judge or other qualitative assessment protocols that could better capture improvements in generation quality.

4. **Limited analysis of the proposed methods.** The paper would benefit from deeper analysis, such as ablations on key design choices (e.g., FOCUS weighting parameters and RePAIR margin/objective components), analysis of how pruning ratio affects degeneration mitigation, and investigation of when the method succeeds or fails (e.g., different decoding strategies or repetition onset contexts).

[1] MiniLLM: Knowledge Distillation of Large Language Models. ICLR 2024

[2] Rethinking Kullback-Leibler Divergence in Knowledge Distillation for Large Language Models. COLING 2024

[3] ToDi: Token-wise Distillation via Fine-Grained Divergence Control. EMNLP 2025

---

> ### Author Rebuttal · Authors · 2026-03-29
>
> We sincerely thank the reviewer for their insightful and constructive feedback. We address the questions below. Due to space constraints, please see the anonymous link for tables and figures (https://anonymous.4open.science/r/exp_figures-B63F)
>
> [1]
>
> The primary objective of our work is to address degeneration, with a particular focus on repetition. Accordingly, we mainly compared our method with prior approaches that are explicitly designed to reduce repetition. As the reviewer suggested, directly comparing with distillation-objective-based methods and examining whether they also help mitigate repetition would provide a clearer perspective on the contribution of our approach.
>
> Regarding MiniLLM, it adopts an RL-based distillation framework where the student generates full sequences and the teacher provides evaluation signals. This approach is computationally demanding, which limits our ability to include it within the rebuttal period. So, we followed the reviewer’s suggestion and compared with ToDi, which is the most closely aligned with FOCUS in terms of token-level KL-based design.
>
> As shown in Table 3 in the linked material, ToDI is not as effective in reducing repetition. However, it achieves better quality compared to the model using FOCUS alone. We believe this is because TODI leverages both forward and reverse KL to more closely match the teacher distribution, whereas FOCUS emphasizes high-confidence regions of the teacher distribution. We will include a more detailed discussion of this aspect in the discussion section.
>
> [2]
>
> We additionally conducted experiments on the Qwen2.5-3B model. To further demonstrate that our method is not tied to a specific pruning approach, we also applied FLAP and shortend-llm to the LLaMA3.2-3B model.
>
> As shown in Table 1 and 2 in the linked material, our method consistently achieves substantial reductions in repetition across all settings. These improvements generalize beyond the LLaMA family to the Qwen model, and further extend to different pruning strategies, as demonstrated with FLAP and shortened-llm.
>
> Regarding the pruning ratios, we focused our experiments on 25%. As noted in prior work and  Appendix table 9, more aggressive pruning tends to result in substantial degradation in generation quality. In such regimes, it becomes increasingly difficult to disentangle repetition-related effects from other degradation issues introduced by excessive pruning. We will clarify this consideration and its implications in the discussion section.
>
>
> [3]
>
> The current evaluation primarily relies on automatic metrics. To address this, we conducted additional evaluations using an LLM-as-a-judge framework. Following the experimental setup in Table 3, we generated Wiki continuation samples using Llama 3 8B and evaluated them with a GPT-120B (reasoning) model as the judge. The evaluation protocol follows prior work: given two candidates (A and B), the judge selects A, B, or TIE. To mitigate positional bias, we randomized the order of A and B and counted inconsistent preferences as TIE.
>
> As shown in Table 4 in the linked material, our method consistently achieves better performance not only on the conventional static metrics but also in LLM-as-a-judge evaluations. We will include more detailed experimental settings and prompt templates in the appendix.
>
> [4]
>
> To provide a more thorough analysis, we conducted additional experiments covering the following aspects. Due to space limitations, please refer to the items below:
>
> Loop persisence analysis (see Reviewer 4jRd, [3])
>
> Loop entry analysis (see Reviewer oM7v, [W1])
>
> CREP threshold sensitivity (see Reviewer oM7v, [W4, Q3])
>
> [Additional]
>
> The evaluation focuses on training-based baselines but does not compare against commonly used decoding-time mitigation methods. As suggested by the reviewer, we investigate whether commonly used decoding strategies can mitigate repetition without additional training, in order to better understand the practical advantages of our approach.
> Specifically, we apply standard decoding-time methods immediately after pruning, including repetition penalty(from 1.0 to 1.3), beam search(n=4), and top-p(p=0.9) sampling, and evaluate them on the Wiki continuation setting.
>
> As shown in Table 5 and 6 in the linked material, repetition penalty can effectively reduce repetition. However, consistent with prior observations, this comes at a substantial cost in text quality, as reflected by a significant drop in MAUVE (compared to 0.73 achieved by our full method). On the other hand, beam search and sampling-based methods would maintain better text quality, but fail to sufficiently suppress repetition. These results suggest that while decoding-time interventions can partially address repetition, they exhibit a clear trade-off between repetition reduction and generation quality, highlighting the practical benefit of training-based approach.

---

> > ### Author Rebuttal · Reviewer_nJHu · 2026-04-05
> >
> > Fully resolved - My concerns have been adequately addressed.

---

### Official Review · Reviewer_4jRd · 2026-03-14

**Soundness:** 3
**Presentation:** 3
**Significance:** 3
**Originality:** 3
**Overall Recommendation:** 4
**Confidence:** 4

**Summary:**

This paper studies a failure mode of pruned large language models: repetitive text degeneration that worsens after pruning even when perplexity or zero-shot accuracy appear mostly preserved. The authors argue that repetition loops can be understood through a token-level decoding dynamics view that separates loop entry risk from loop persistence, with persistence governed by “escape mass” among plausible alternatives inside the nucleus set under top-p sampling. Based on this framing, they propose two post-pruning training objectives: FOCUS, a reweighted knowledge distillation loss that emphasizes high-confidence teacher regions to reduce leakage into teacher-suppressed tokens, and RePAIR, a pairwise margin objective built from repetition-onset continuations that encourages non-repetitive alternatives. Experiments on WikiText-103 continuation and Self-Instruct generation show large reductions in CREP and improvements in MAUVE/EAD1/BERTScore, with some perplexity increase.

**Compliance With Llm Reviewing Policy:**

Affirmed.

**Final Justification:**

See rebuttal comment.

**Key Questions For Authors:**

Please see concerns above.

**Limitations:**

Yes.

**Strengths And Weaknesses:**

The paper is written clearly and both the problem and the approach are motivated well. I enjoyed reading the paper. There are nice empirical results showing that focus + repair reduce degeneration, as well as interesting side analyses like replacing the two tokens at the onset of the repetition loop.

Some concerns/questions:

1. In terms of baselines, how does the proposed method compare with reverse KL distillation, which seeks to solve the same mode-covering issue (e.g., Gu et al. 2024 which is cited in the paper)? That seems to be one of the most direct comparisons.

2. Experiments aren't that broad: they only evaluate on Llama models, with one type of pruning (up to 45%).

3. I found the theoretical parts interesting but a bit speculative, in the sense that there aren't really direct experiments linking, e.g., the focus method to loop entry. Perhaps the framing could be adjusted. I also didn't follow some of the explanations, e.g., "As a result, moderate reductions in ρ¯ can suppress long repetition runs even when entry events are not fully eliminated. This explains why changing only a few high-probability tokens near onset can produce a large reduction in CREP in Table 2." --> why does this imply that the change in loop persistence is the cause here? I'd have expected that changing two tokens near the onset would change the loop entry instead.

---

> ### Author Rebuttal · Authors · 2026-03-29
>
> We sincerely thank the reviewer for their thoughtful and detailed feedback. We address the questions below. Due to space constraints, please see the anonymous link for materials (https://anonymous.4open.science/r/exp_figures-B63F)
>
> [1]
>
> The primary goal of our paper is to mitigate degeneration, particularly repetition. Accordingly, we focused our comparisons on methods that directly address repetition (e.g., DiTTO, unlikelihood training, etc.). As the reviewer pointed out, comparing with other KL-based approaches would be helpful to better highlight the necessity and contribution of FOCUS.
>
> Specifically, reverse KL can be understood as adjusting the student distribution based on the teacher probabilities over tokens that the student already assigns probability to. As a result, tokens that receive negligible or zero probability under the student are not encouraged, even if the teacher assigns them non-trivial probability mass, leading to a mode-seeking behavior.
>
> It has been observed that reverse-KL-based objectives tend to exhibit reduced generation diversity (Gu et al., 2024), consistent with prior observations (e.g., [CCF+20]). Intuitively, even if the teacher assigns probability to semantically valid alternative tokens, the student may fail to explore them if they are not already supported by its own distribution. This can suppress potentially useful alternatives during generation.
>
> In contrast, our method follows a forward-KL-based distillation with a reweighted teacher distribution (Appendix B), which explicitly reshapes the target distribution to suppress repetition-inducing structures. Rather than relying on the student’s current support, our approach directly modifies the teacher signal to control the balance between dominant and alternative tokens, leading to improved repetition behavior (Please refer results in [3]).
>
> As a representative example, we compared FOCUS with the hybrid-KL (ToDi) method suggested in our reviews for the LLaMA3-8B model similarly as done in the main paper. As shown in Table 3 in the link, ToDi is not as effective in reducing repetition. However, it achieves better quality compared to the model using FOCUS alone. We believe this is because ToDi leverages both forward and reverse KL to encourage broader mode coverage of the teacher distribution, whereas FOCUS emphasizes high-confidence regions of the teacher distribution. Since the direct goal of other reverse KL methods such as MiniLLM is not suppressing degeneration, we expect similar results; due to the limited rebuttal period and the high computational cost of MiniLLM, we will include more detailed exploration in the final version.
>
> [2]
>
> We additionally evaluate our method across different pruning strategies and model architectures. Please refer to Reviewer nJHu [3] for details.
>
> Regarding the pruning ratios, we focused our experiments on 25%. As noted in prior work, more aggressive pruning tends to result in substantial degradation in generation quality. In such regimes, it becomes increasingly difficult to disentangle repetition-related effects from other degradation issues. Consistent with this, as shown in Appendix Table 9, we observe that while higher pruning ratios further reduce repetition, they also lead to a rapid degradation in generation quality. We will clarify this consideration and its implications in the discussion section.
>
> [3]
>
> To more explictly present how the proposed method affects the loop persistence, we conducted additional empirical validation experiments to more directly support our design.
>
> We construct synthetic repetition by taking 1000 consecutive sentences from WikiText, using the first as a prefix and repeating the second sentence 20 times. We then compute (i) a geometric mean probability score as a proxy for the probability of repetition,  $\exp\left(\frac{1}{|\mathcal{A}(c)|}\sum_{t \in \mathcal{A}(c)} \log p(t \mid c)\right)$, where $\mathcal{A}(c)$ denotes the set of tokens along the repetition path. Second, we directly compute Eq. (9) in Section 3.3 of the main paper. Specifically, within the repeated sentence, tokens along the repetition path are treated as $a(c)$, while the remaining tokens are treated as $e(c)$, and we compute $\rho(c) = \frac{a(c)}{a(c) + e(c)}$. As shown in Figures 1 and 2 in the link, our method reduces both loop mass and the persistence proxy, making it more likely to escape and reduce repetition loops. (We also did loop-entry experiment. Please see Reviewer oM7v, [W1])
>
> For the explaination,  we also agree that modifying tokens near the onset can be interpreted as affecting loop entry, as these tokens influence whether generation enters a repetitive pattern. At the same time, onset tokens define the initial state of loop dynamics and can affect persistence through the balance between continuation and escape probabilities as shown in Figure 1,2 and 3 in the link. We will revise the text to more clearly explain them.

---

> > ### Author Rebuttal · Reviewer_4jRd · 2026-04-03
> >
> > Thanks to the authors for the rebuttal. My questions have been addressed (except for the reverse KL experiment which I acknowledge is hard to run in the rebuttal period). I will maintain my score.

---

### Decision · Program_Chairs · 2026-04-30

**Decision:**

Accept (spotlight)

**Comment:**

**Summary:** This paper addresses the critical issue of repetition-loop degeneration in pruned Large Language Models (LLMs). The authors analyze this phenomenon through a token-level decoding dynamics perspective, effectively decomposing the failure mode into two factors: loop entry risk and loop persistence. To mitigate these issues, they introduce two post-pruning training objectives. FOCUS is a reweighted knowledge distillation loss designed to suppress probability leakage into tokens suppressed by the teacher model, while RePAIR utilizes a pairwise margin loss at repetition onsets to encourage the model to explore escape alternatives. Experimental results across open-ended generation and instruction-following tasks demonstrate that these methods significantly reduce repetition while maintaining or improving generation quality.

**Summary of Reviews:** The reviewers reached a strong consensus with an average score of 4.5. A primary initial concern was the limited experimental coverage, specifically the absence of competitive baselines such as reverse-KL, ToDi, or MiniLLM. In response, the authors provided a thorough rebuttal that included direct comparisons with the ToDi method, demonstrating that while ToDi achieves high quality, it is less effective at suppressing repetition than the proposed approach. Furthermore, the rebuttal expanded the evaluation to the Qwen model family and additional pruning methods like FLAP, which effectively resolved concerns regarding the generalizability of the findings.

**Assessment:** The submission identifies a vital failure mode where pruned models appear performant on perplexity benchmarks but degenerate during practical open-ended generation. The proposed framework provides a robust foundation for the two training objectives, which are both intuitive and easy to integrate into existing pipelines. It is worth noting that while the method results in a slight increase in perplexity, the authors demonstrated that this does not compromise factual accuracy as measured by TruthfulQA. Additionally, the proposed training-based approach was shown to be superior to standard decoding-time interventions, such as repetition penalties, which often significantly degrade overall text quality.Based on the discussion, the final version should temper the initial strong claim of a strict one-to-one functional mapping between FOCUS/entry and RePAIR/persistence, as the evidence suggests these mechanisms are more intertwined. Nevertheless, with all reviewers positive and a clear demonstration of empirical effectiveness, this work provides a significant contribution to the field of model compression.